# TIME SERIES SALIENCY MAPS:
# EXPLAINING MODELS ACROSS MULTIPLE DOMAINS

## ABSTRACT

Traditional saliency map methods, popularized in computer vision, highlight individual points (pixels) of the input that contribute the most to the model's output. However, in time-series, they offer limited insights, as semantically meaningful features are often found in other domains. We introduce Cross-domain Integrated Gradients, a generalization of Integrated Gradients. Our method enables feature attributions in any domain that can be formulated as an invertible, differentiable transformation of the time domain. Crucially, our derivation extends the original Integrated Gradients into the complex domain, enabling frequency-based attributions. We provide the necessary theoretical guarantees, namely, path independence and completeness. Our approach reveals interpretable, problem-specific attributions that time-domain methods cannot capture in three real-world tasks: wearable sensor heart rate extraction, electroencephalography-based seizure detection, and zero-shot time-series forecasting. We release an open-source Tensorflow/PyTorch library to enable plug-and-play cross-domain explainability for time-series models. These results demonstrate the ability of cross-domain integrated gradients to provide semantically meaningful insights into time series models that are impossible to achieve with traditional time-domain saliency.

## 1 INTRODUCTION

Saliency maps are visual tools for explaining deep learning models Selvaraju et al. (2017). Popularized in computer vision, they highlight input points that contribute the most to the model's output Gupta et al. (2022). For images, the original input domain, pixels, aligns naturally with human perception, as neighboring pixels form coherent objects that are understood by human vision. This makes pixel-level saliency intuitive and semantically meaningful. Similarly, in natural language processing Li et al. (2016), word-level attributions can be informative, as words inherently bear semantic meaning.

In contrast, in time series, this intuition breaks down Schröder et al. (2023b;a). In the time domain, groups of temporally adjacent points, i.e., the equivalent of a pixel, do not necessarily form intuitive *concepts*. Instead, such *concepts* are found in intricate interactions between points, linking them to higher-level abstractions such as oscillating frequency patterns or statistically independent formations. As a consequence, highlighting individual time points does not provide meaningful insight into the behavior of the model.

Signal processing practice has long faced this challenge, where signal interpretation generally relies on the decomposition of the original signal into structured *components*. Through transformations, the original time domain is mapped to the component domain, capturing the higher-level interactions and linking the input to semantically meaningful concepts. The choice of decomposition and component domain depends on the nature of the signals and the task. For example, the Fourier transform decomposes the original signal into sinusoidal oscillations Bracewell (1989), while the Independent Component Analysis (ICA) decomposes the signal into statistically independent components Lee and Lee (1998). Such transformations map the time signals into structured, semantically rich domains, providing more intuitive interpretations of the signal's contents.

In their work, Schröder et al. (2023b;a) demonstrates that time domain saliency often fails when labels depend on latent features like frequency components and argues that visual explanations of time-series models should be expressed in this latent space rather than solely in the time domain. Building on their empirical observations, we argue that saliency methods should be capable of generating

attributions in interpretable domains, even when the model processes time points. This motivates the need for saliency map tools that can visualize feature importance across multiple domains.

To address this, we develop Cross-domain Integrated Gradients, a novel method for visualizing feature importance across multiple domains. Based on the principles of IG Sundararajan et al. (2017), we derive the formulas, axioms, and proofs required to apply IG across domains. We validate our method by following the exact same steps as IG Sundararajan et al. (2017). We show that cross-domain IG maintains the Completeness property, hence satisfying *Sensitivity* and *Implementation Invariance*. We apply our method to real-world time-series models and applications, demonstrating that descriptive domains can be very powerful in understanding model behavior.

In this work, we introduce the following novel contributions:

- We propose a generalization of the Integrated Gradients that enables cross-domain explainability for any invertible transformation, including non-linear ones.

- We derive a generalization of the Integrated Gradients for real-valued functions with a complex domain, enabling the generation of frequency-domain saliency maps.

- We demonstrate how different domains allow for a better understanding of model behavior on time-series data.

- We release an open-source Python library compatible with `tensorflow` and `pytorch` for cross-domain time series explainability. The code for reproducing the results of this paper, along with the library, is available in the supplemental material. Upon acceptance, we will also include the corresponding open-source Github links.

## 2 RELATED WORKS

**Saliency map interpretation.** Saliency maps, as a means of interpreting the behavior of the model, have been popularized in computer vision. These methods generate an output that maps each individual input pixel to a significance score. Several methods have been proposed for this mapping. Activation-based methods, such as GradCAM Selvaraju et al. (2017) and later variations Chattopadhay et al. (2018), generate saliency based on deep layer activations. Gradient-based methods, such as Integrated Gradients (IG) Sundararajan et al. (2017); Kapishnikov et al. (2021), generate significance scores by using the model's output gradients with respect to its inputs. Similarly, Layer-wise Relevance Propagation (LRP) methods Bach et al. (2015) propose rules to propagate the model output backwards by distributing the overall output among individual input features.

**Time domain explainability.** Saliency map methods have been applied to time series applications, either by direct application of computer vision-derived methods Jahmunah et al. (2022); Tao et al. (2024) or by developing dedicated time series saliency approaches Queen et al. (2023); Liu et al. (2024). To streamline comparisons between time-domain interpretability, Ismail et al. (2020) proposed an extensive synthetic, multi-channel benchmark. In all cases, these approaches focus on identifying significant regions of the time-domain input which contribute the most to the model's output. Such regions of interest are events that trigger the model's output.

**Cross domain interpretability.** The current time domain saliency methods have limitations, as highlighted time points do not always explain the underlying mechanisms Theissler et al. (2022). Schröder et al. (2023b;a) formalised this time-domain limitation, arguing that saliency maps need to be generated in the latent-space, where the useful information lives, providing empirical evidence of time-domain saliency map failure modes. Furthermore, Chung et al. (2024) demonstrate that such methods are not robust to frequency perturbations. These limitations diminish the explanatory power of the generated saliency map. To address this issue, they proposed a perturbation method in the time-frequency domain, attributing the model output to time-frequency components. However, frequency perturbations can strongly affect model performance and, therefore, explainability due to out-of-distribution effects Sundararajan et al. (2017). Similarly, Vielhaben et al. Vielhaben et al. (2024) proposed the *virtual inspection layer* placed after the model input to transform the saliency map of the time domain to the frequency and time frequency domains, proposing dedicated relevance propagation rules for the frequency transform. Brüsch et al. (2024) and Brüsch et al. (2025) proposed

masking-based saliency methods, expressing feature significance in the frequency and time-frequency domains.

**Saliency map evaluation.** Evaluating saliency maps is not a trivial task. A major challenge lies in disentangling saliency map errors from model errors Kim et al. (2021); Akhavan Rahnama (2023), complicating validation through comparison with ground truth saliency. Sundararajan et al. (2017) proposes solving this by relying on a set of desirable axioms, bypassing the necessity for empirical evaluations. Validation based on insert / deletion is another approach Hama et al. (2023); Ismail et al. (2020). These methods empirically evaluate the effect of removing/retaining the most important input features, reinforcing trust in the saliency map method under examination.

Despite progress in time-series saliency, existing methods (i) operate solely in the time domain, (ii) rely on perturbation-based attributions only in the frequency domain, or (iii) require transform-specific hand-crafted relevance-propagation rules valid only in the frequency domain. In contrast, our work provides a principled generalization of Integrated Gradients that supports any invertible, differentiable transform, including complex-valued domains, while preserving axiomatic properties and enabling semantically meaningful attributions across diverse time series applications.

## 3 PRELIMINARIES

### 3.1 PROBLEM STATEMENT AND MOTIVATION

We consider a function $f : \mathcal{D}_s \to \mathbb{R}$ representing a deep learning model. The input $\boldsymbol{x} \in \mathcal{D}_s$ is constructed from a continuous-time signal $x(t) \in \mathbb{R}$ after discretizing it at a sampling frequency $f_s$ [Hz] and considering a window of length $L$ seconds: $\boldsymbol{x} = [x_0, ..., x_{n-1}], n = f_s \cdot L$. Now consider a transform $T : \mathcal{D}_S \to \mathcal{D}_T$ that maps the original time domain to a semantically rich explanation target domain $\mathcal{D}_T$. Our task is to construct an informative saliency map that assigns a significance score to each characteristic $z_i = T(\boldsymbol{x})_i$ in the explanation domain.

Saliency maps developed in computer vision applications, and in particular IG, provide explanations in the same domain as the model's input, that is, $\mathcal{D}_T = \mathcal{D}_S$. Applying these methods to time-series models results in maps expressed in the time domain.

We summarise the empirical conclusions of relevant works Schröder et al. (2023b;a); Theissler et al. (2022); Vielhaben et al. (2024), in Proposition 1.

**Proposition 1.** *The time domain is not always informative in explaining $f$.*

We provide further analytically tractable evidence in support of Proposition 1 through our example in Section 3.2 which is in line with the empirical synthetic experiments of Schröder et al. (2023b). Although this example focuses on the frequency domain, our derivation is transform-agnostic (Section 4) and we expand to more domains in Section 5, providing real-world cases after formally defining our method.

### 3.2 TIME DOMAIN EXPLANATION LIMITATIONS

Consider that the input $\boldsymbol{x}$ is sampled from the signals $x(t) = cos(2\pi\xi t + \phi)$. In this setup, there are two classes of samples depending on the oscillating frequency $\xi$:

$$y = \begin{cases} 1, & \xi \sim \mathcal{N}(1.0, 0.5) \\ 2, & \xi \sim \mathcal{N}(4.0, 0.5) \end{cases} \tag{1}$$

We design a classifier $f$ to distinguish between these two classes. We opt to manually construct $f$ so that we have full mechanistic understanding of its inner workings. We choose a CNN architecture composed of a single convolutional layer with two channels followed by a ReLU activation and global average pooling $f(\boldsymbol{x}) = AvgPool\left(ReLU(\boldsymbol{w} * \boldsymbol{x})\right)$. The kernel of the first channel is a low-pass filter (cutoff at $2.5Hz$), while the second channel kernel is a high-pass filter with the same cutoff (see Figure 1).

Ideally, the model should be fully explained by describing its inner mechanisms. In this particular scenario, we have designed $f$ for this purpose; hence, a formal detailed explanation is available.

**Mechanistic Interpretation 1.** *Convolutional channel $i$ allows only frequencies of class $i$ to pass through the output; otherwise, the channel's output is almost zero, not activating. The ReLU and Average Pooling mechanism extract the amplitude of the signal Kechris et al. (2024a). Hence, the channel $i$ of the model output is only active when samples from class $i$ are processed, leading to the correct classification of the input.*

That depth of model understanding is not easily available in larger models, which have been trained on samples. Hence, saliency maps are often used as a proxy. We provide IG explanations of the model $f$ for samples from both classes, expressed in the time and frequency domains (Figure 1). Although time points are periodically highlighted as *more important*, it is not exactly clear how this input influences the model towards producing its output.

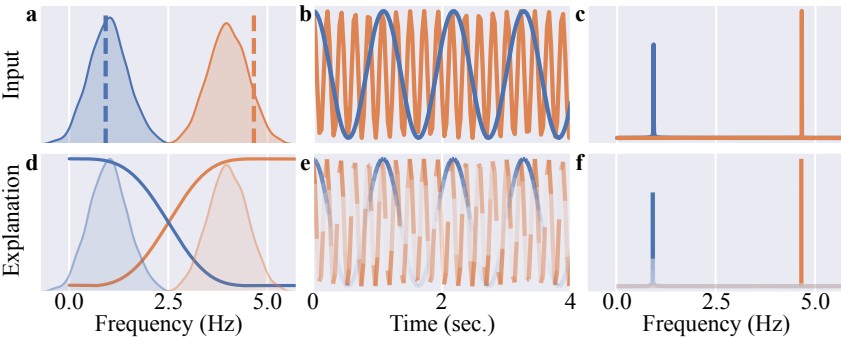

Figure 1: **Mechanistic interpretation along with Time and Frequency domain saliency maps.** (a) Distributions of the main frequency, $\xi$, for classes **one** and **two**. We sample one input for each class (vertical dashed lines) for which we generate the saliency maps. (b) These two sampled inputs presented in the time and (c) frequency domains. (d) Illustration of the Mechanistic Interpretation1. We plot the frequency response for the **first** and **second** channels of the CNN. The sample distributions (a) are also overlayed. (e) Saliency maps expressed in the time and (f) frequency domains.

In contrast, a saliency map expressed in the frequency domain, which we introduce in Section 4, highlights the frequency components that contribute to the final output: for the samples of class one, only the 1 Hz component contributes to the model's output, and accordingly, for class two, the 4 Hz component. Here, this saliency map is much more interpretable. It provides useful information and better aligns with the mechanistic understanding (Mechanistic Interpretation 1) of this model. In Section 4, we show analytically that the frequency-expressed IG, for the data distribution and model of this example, is directly linked to its mechanistic explanation.

### 3.3 INTEGRATED GRADIENTS

To explain the output of a model $f$ on an input $\boldsymbol{x}$ with a baseline $\hat{\boldsymbol{x}} \in \mathbb{R}^n$, IG generates a saliency map as Sundararajan et al. (2017):

$$IG_i(\boldsymbol{x}) = (x_i - \hat{x}_i) \int_0^1 \frac{\partial f}{\partial x_i}\bigg|_{\boldsymbol{x}' + t \cdot (\boldsymbol{x} - \hat{\boldsymbol{x}})} dt \qquad (2)$$

with each element $IG_i(x)$ of the map corresponding to the significance of the input feature $x_i$: saliency is expressed in the same domain as the input. The IG definition relies on two key points from the theory of integrals over differential forms: the line integral definition and Stokes' theorem.

**Line integral definition.** The IG can be derived from the definition of the integral of the differential form $df$ along the line $\boldsymbol{\gamma}(t) = \hat{\boldsymbol{x}} + t(\boldsymbol{x} - \hat{\boldsymbol{x}})$:

$$\int_\gamma df = \int \boldsymbol{\gamma}^* df = \int_0^1 \sum_{i=0}^N \frac{\partial f}{\partial x_i} \gamma_i'(t) dt = \sum_{i=0}^N \int_0^1 \frac{\partial f}{\partial x_i} \gamma_i'(t) dt = \sum_{i=0}^N (x_i - \hat{x}_i) \int_0^1 \frac{\partial f}{\partial x_i} dt \qquad (3)$$

where $\boldsymbol{\gamma}^* df$ is the pullback of $df$ by $\boldsymbol{\gamma}$: $\boldsymbol{\gamma}^* df = \sum_{i=0}^N \frac{\partial f}{\partial x_i} \gamma_i'(t) dt$ Do Carmo (1998). Each individual element of the IG map $IG_i(\boldsymbol{x})$ corresponds to each element of the last sum of eq. 3.

**Stoke's Theorem.** The *Completeness* axiom of the IG Sundararajan et al. (2017): $f(\boldsymbol{x}) - f(\hat{\boldsymbol{x}}) = \sum IG_i$ is a consequence of the Stokes' Theorem for the case of integral of 1-form: $\int_\gamma df = \int_{\partial\gamma} f = f(\boldsymbol{x}) - f(\hat{\boldsymbol{x}})$, which guarantees path independence: the value of the integral is only dependent on the first and last points of the path, not the path itself.

## 3.4 SALIENCY MAPS EVALUATIONS

Saliency map evaluation is challenging (Section 2), therefore we adopt a broad, complementary validation protocol that triangulates evidence from theory, controlled experiments, qualitative sanity checks, and dataset-level stress tests:

1. **Axiomatic soundness.** We show that Cross-domain IG maintains the Completeness property, hence satisfying *Sensitivity* and *Implementation Invariance* Sundararajan et al. (2017).

2. **Mechanistic alignment.** Based on the example in Section 3.2, we theoretically show that cross-domain IG can align with the model's internal mechanisms - when the target domain is appropriate (Section 4.2).

3. **Qualitative applications.** We show representative examples in Section 5 that demonstrate the full Cross-Domain IG workflow and how it can uncover data/model insights.

4. **Quantitative sufficiency/necessity.** We conduct insertion-deletion tests on real-world time-series datasets (Appendix G).

5. **Comparison with SoTA.** We analytically compare our method to Vielhaben et al. (2024) (Appendix E). We also empirically compare to Brüsch et al. (2024) (Appendix F).

## 4 METHODS

In this section, we define Cross-Domain IG (Section 4.1), and derive it based on the IG principles from Section 3.3. We then analyse it in the complex frequency domain using a simple yet representative convolutional network, highlighting its relation to the network's properties (Section 4.2). This analysis also provides theoretical grounding for the connection between frequency-domain IG and the Mechanistic Interpretation discussed in Section 3.2. Finally, we detail the implementation of our method.

### 4.1 CROSS-DOMAIN IG DERIVATION

Let $f : \mathcal{D}_s \to \mathbb{R}$ be a deep neural network operating on a domain $\mathcal{D}_s \subseteq \mathbb{R}^n$. Also, denote $\boldsymbol{x}, \hat{\boldsymbol{x}} \in \mathcal{D}_s$ the input and baseline samples, respectively, as defined by the IG method. We introduce an invertible, differentiable transformation $T : \mathcal{D}_S \to \mathcal{D}_T$ and its inverse $T^{-1}$, which is also differentiable, with $\boldsymbol{z} = T(\boldsymbol{x})$, $\boldsymbol{x} = T^{-1}(\boldsymbol{z})$, and $\mathcal{D}_T \subseteq \mathbb{C}^m$. The cross-domain IG generates the saliency map for $f$, attributing the difference $f(\boldsymbol{x}) - f(\hat{\boldsymbol{x}})$ to the features $\boldsymbol{z}$, expressed in $\mathcal{D}_T$. To define Cross-domain Integrated Gradients, we consider the path integral of model gradients over the transformed feature space:

**Definition 4.1** (Cross-domain Integrated Gradients). *Given a model* $f : \mathcal{D}_s \to \mathbb{R}$, *a transform* $T : \mathcal{D}_S \to \mathcal{D}_T$ *and its inverse* $T^{-1}$, *input and baseline samples* $\boldsymbol{x}, \hat{\boldsymbol{x}} \in \mathcal{D}_s$ *and* $\boldsymbol{\gamma}(t)$ *the line from* $\boldsymbol{z} = T(\boldsymbol{x})$ *to* $\hat{\boldsymbol{z}} = T(\hat{\boldsymbol{x}})$ *the Cross-Domain IG is defined as:*

$$IG_i^{\mathcal{D}_T}(\boldsymbol{z}) = 2\int_0^1 \mathrm{Re}\left\{ \left.\frac{\partial(f \circ T^{-1})}{\partial z_i}\right|_{\boldsymbol{\gamma}(t)} \cdot (z_i - \hat{z}_i)\right\} dt \tag{4}$$

Note that the original IG, eq. 2, and $IG^{\mathcal{D}_T}$ explain the exact same functionality since $f(\boldsymbol{x})$ and $(f \circ T^{-1})(\boldsymbol{z})$ are equivalent. However, their output saliency maps are expressed in different domains. We now derive Definition 4.1 from the first principles of the original IG method, Section 3.3.

**Derivation sketch.** The original IG is only defined for real inputs. To enable complex-valued transformations, such as the Fourier transform, we extend IG for real-valued functions $g$ with complex inputs $\boldsymbol{z}$, referred to as *Complex IG*. Our derivation builds on the two key points in Section 3.1:

1. **Line integral definition.** We begin our derivation by defining a function $u$ that is equivalent to $g(\boldsymbol{z})$. Just like in the case of real inputs, eq. 2, we elaborate on the line integral $\int_{\gamma} du$. The end goal is to end up with a sum of integrals $\sum_i \int ...dt$ similar to eq. 3. In the final step, each IG element is defined as the corresponding integral term of the final sum, $\int ...dt$.

2. **Stokes' Theorem.** We define $u$ and derive complex IG to ensure path independence and satisfy the *Completeness axiom*, which may fail for functions of several complex variables Lebl (2019). To this end, we first state and prove Lemma 4.1 as an intermediate result. Using Lemma 4.1, we then derive Definition 4.1 using Wirtinger calculus.

**Lemma 4.1.** *Let $g : \mathbb{C}^n \to \mathbb{R}$, $\boldsymbol{z} = \boldsymbol{p} + j\boldsymbol{q}$, with $\boldsymbol{p}, \boldsymbol{q} \in \mathbb{R}^N$, $\boldsymbol{\gamma}(t) = \hat{\boldsymbol{z}} + t(\boldsymbol{z} - \hat{\boldsymbol{z}}), t \in [0, 1]$ the line from the baseline point $\hat{\boldsymbol{z}}$ to the input point $\boldsymbol{z}$ and $\boldsymbol{n}(t) = \text{Re}\{\boldsymbol{\gamma}(t)\}$ and $\boldsymbol{m}(t) = \text{Im}\{\boldsymbol{\gamma}(t)\}$, $\boldsymbol{n}(t), \boldsymbol{m}(t) \in \mathbb{R}^n$. Then the IG of $g$ in $\boldsymbol{z}$ is given by:*

$$IG_i^{\mathbb{C}^n}(\boldsymbol{z}) = \int_0^1 \left( \frac{\partial g}{\partial p_i} n_i'(t) + \frac{\partial g}{\partial q_i} m_i'(t) \right) dt \tag{5}$$

A detailed proof of Lemma 4.1 can be found in Appendix B. From Lemma 4.1, and considering $g(\boldsymbol{z}) = f\left(T^{-1}(\boldsymbol{z})\right)$ and the complex differential form Range (1998) $dg = \partial g + \overline{\partial} g$ we can write the complex integrated gradient definition as:

$$IG_i^{\mathbb{C}^n} = 2 \int_0^1 \text{Re}\left\{ \frac{\partial g}{\partial z_i} \gamma_i'(t) \right\} dt \tag{6}$$

The complete derivation can be found in Appendix C. Notice that Cross-domain IG maintains the *Completeness* property since $\int_{\gamma} du = u(\boldsymbol{a}(1)) - u(\boldsymbol{a}(0)) = g(\boldsymbol{z}) - g(\hat{\boldsymbol{z}}) = f(\boldsymbol{x}) - f(\hat{\boldsymbol{x}})$, where $u : \mathbb{R}^{2n} \to \mathbb{R}$ s.t. $g(\boldsymbol{p} + j\boldsymbol{q}) = u([\boldsymbol{p}, \boldsymbol{q}])$ and $\boldsymbol{a} = [\boldsymbol{n}, \boldsymbol{m}]$.

**Remark 1.** *Although definition 4.1 defines a linear path of integration, in our derivation, Eq. 6, the path of integration is a general curve $\boldsymbol{\gamma}(t)$. This enables incorporating into cross-domain IG alternative integration paths/methods to reduce sensitivity to noise Yang et al. (2023); Kapishnikov et al. (2021).*

**Cross-Domain IG for real-valued inputs.** If $g$ processes real-valued inputs, then eq. 6 is equivalent to eq. 2: since $g(\boldsymbol{z}) = g(\boldsymbol{p} + j0)$, $\partial g / \partial q = 0$, $\partial g / \partial z = (1/2) \partial g / \partial p$. Thus, if $\mathcal{D}_T \subseteq \mathbb{R}^n$ the cross-domain IG can equivalently be expressed as $IG_i^{\mathcal{D}_T}(\boldsymbol{z}) = (z_i - \hat{z}_i) \int \frac{\partial (f \circ T^{-1})}{\partial z_i} dt$.

**Remark 2.** *In IG Sundararajan et al. (2017), the baseline $\hat{\boldsymbol{x}}$ is defined as the point without information about the original model inference. The authors argued that most deep networks admit such a neutral input. For cross-domain IG, if $\hat{\boldsymbol{x}}$ exists, and $T$ is invertible, then $\hat{\boldsymbol{z}}$ is trivially defined. Crucially, cross-domain IG enables baselines that were not easily defined, e.g., filtering specific components from $\boldsymbol{x}$ to form $\hat{\boldsymbol{z}}$.*

## 4.2 COMPLEX IG ON A SIMPLE MODEL

Adebayo et al. (2018) analytically studies a minimal single-layer convolutional network, demonstrating that IG can collapse into an *edge detector*, producing misleading saliency maps. Although this exposes a failure mode of the IG in the input domain, we show that Complex-IG faithfully reflects the inner mechanisms of a simple convolutional network in the frequency domain. In direct parallel, we derive a closed-form link between the complex IG saliency map of a CNN and the frequency response of its filters. Building on the example in Section 3.2, we work on a simple CNN and prove that Complex-IG highlights each filter's gain at its corresponding input frequency.

Let $f$ be a convolutional neural network composed of a single convolutional layer (1 channel) followed by a ReLU operation and Global Average Pooling: $f(\boldsymbol{x}) = AvgPool\left(ReLU(\boldsymbol{w} * \boldsymbol{x})\right)$. We begin with the case in which $f$ processes windows sampled from single-component sinusoidal signals $x(t) = a_j \cdot cos(2\pi\xi_j t + \phi)$, $a_j > 0$. Then, the output $f(\boldsymbol{x})$ is Kechris et al. (2024a): $f(\boldsymbol{x}) = \frac{a_j b_j}{\pi}$, with $b_i$ the amplification of the filter $\boldsymbol{w}$ at frequency $\xi_i Hz$: $b_i = \|\sum_n w_n e^{-2\pi\xi_i n}\|$. We employ the Complex IG method on $f$ with baseline input $\hat{\boldsymbol{x}} = \boldsymbol{0}$, $f(\boldsymbol{0}) = 0$. This yields $IG_i^{\mathbb{C}^n} = 0$, $\forall i \neq j$ and $\sum_i IG_i^{\mathbb{C}^n} = f(\boldsymbol{x}) - f(\hat{\boldsymbol{x}})$. Thus,

$$IG_j^{\mathbb{C}^n} = f(\boldsymbol{x}) = \frac{a_j b_j}{\pi} \tag{7}$$

This links $IG_j^{\mathbb{C}^n}$ to the output frequency content $a_j b_j$ and, by extension, to the convolutional filter's frequency response. An example for the model of Section 3.2 is presented in Figure 5 (Appendix D).

### 4.3 Implementation

Autograd (pytorch / tensorflow) allows for automatic differentiation with complex variables using Wirtinger calculus Kreutz-Delgado (2009). Thus, the complex IG can be directly approximated by autograd, using Definition 4.1 or Lemma 4.1, with the detail that Autograd (in both libraries) calculates the conjugate of the complex partial derivative. For the integral calculation, we use a summation approximation similar to Sundararajan et al. (2017). The algorithms for estimating cross-domain IG for the case of $\mathcal{D}_T \subseteq \mathbb{R}^n$ and the two implementations on $\mathcal{D}_T \subseteq \mathbb{C}^n$ (Lemma 4.1 and Definition 4.1) are presented in Algorithms 1 and 2, 3 in the appendix, respectively.

**Remark 3.** *The numerical approximation of the integral in Definition 4.1 requires multiple differentiations, which increases computational overhead. Although the original IG also suffers from similar overhead, our method requires an additional step due to the inverse transform step (see line 9 in Algorithm 3 in the Appendix).*

## 5 Applications

We deploy cross-domain IG in a range of time series applications and models. We selected applications on all three main time-series tasks: regression (section 5.1), classification (section 5.2) and forecasting (Section 5.3). In all three cases, the models are trained to infer on inputs in the time domain. For each application, we first study the properties of the input signal from a signal processing perspective. We then define an interpretability task: *what do we want to learn about our model's behavior through a saliency map?* Based on this domain knowledge and interpretability task, we select an appropriate explanation space yielding semantically meaningful saliency maps. We conclude each example with a remark on actionable insights based on cross-domain attributions. Time-Domain IG attributions of these examples can be found in Appendix H, and additional examples in Appendix I. We also perform feature insertion/deletion evaluation in Appendix G.

### 5.1 Heart rate extraction from physiological signals

We use the KID-PPG Kechris et al. (2024b), a deep convolutional model with attention, to extract heart rate (HR) from photoplethysmography (PPG) signals collected from a wrist-worn wearable device. We use signals from the PPGDalia dataset Reiss et al. (2019). For a time window small enough for the HR frequency, $\xi_{hr}$, to be considered constant, a clean PPG signal can be modeled as Kechris et al. (2024b): $x(t) = a_1 cos(2\pi \cdot \xi_{hr} \cdot t + \phi) + a_2 cos(2\pi \cdot (2\xi_{hr}) \cdot t + \phi)$, with $a_1 > a_2$. However, external signals are also usually present in PPG recordings Reiss et al. (2019); Kechris et al. (2024b). These *interferences* are not created by the heart and are preventing the model from making accurate HR inferences.

**Remark 4.** *KID-PPG processes PPG signals that contain both heart-related components and external interference. A trustworthy model should base the inferred heart rate on heart-related signals only, filtering out all other sources of noise.*

**Interpretability task.** Given a PPG sample and KID-PPG's HR inference, determine whether the model is focusing on heart-related information or external interference.

**Problem-specific transformation.** Since our understanding of this application is mostly frequency-based, we have selected the frequency domain, using the Fourier transform, as the explanation target domain. Hence, the frequency-domain IG highlights individual frequencies as being important to the final model inference. This allows us to investigate whether the HR inference is produced by components related to the heart or by external interference.

An illustration of two PPG inputs and the corresponding frequency-domain IGs is presented in Figure 2. The frequency IG identifies samples in which the model infers heart rate from external interference, thus limiting the reliability of the model's output.

**Remark 5.** *Frequency-domain IG highlights whether KID-PPG inference is **trustworthy** (based on heart oscillations) or **spurious** (based on motion-induced artifacts).*

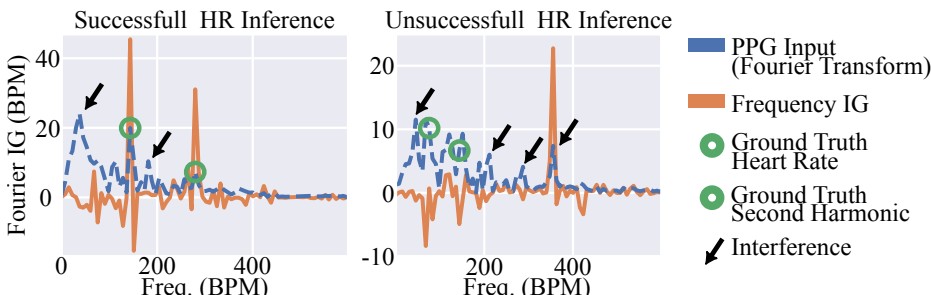

Figure 2: **Frequency-domain IG on heart rate inference model.** The **PPG** signal includes components from the **heart rate** and other components attributed to external interference (denoted with arrows), e.g. motion. **Left:** Sample with a small inference error 0.93 beats-per-minute (BPM). The **IG** highlights the two heart components located at $hr$ and $2 \cdot hr$ (second harmonic), with more weight given to the actual heart rate frequency. **Right:** **PPG** sample with high inference error (26.78 BPM). **IG** coefficients highlight frequency components which are not related to the heart.

## 5.2 ELECTROENCEPHALOGRAPHY-BASED EPILEPTIC SEIZURE DETECTION

We use the `zhu-transformer` Zhu and Wang (2023), which performs seizure detection on scalp-electroencephalography (EEG). We analyze a recording from the Physionet Siena Scalp EEG Database v1.0.0 Detti (2020); Detti et al. (2020); Goldberger et al. (2000). In EEG a single channel captures the electrical activity of multiple *sources*: e.g., epileptic activity, muscle interference, or electrical noise.

**Remark 6.** *A seizure classification model processes the aggregated activity of all sources in the EEG. The model should isolate only the epileptic activity, filtering out all others, to reach a trustworthy inference.*

**Interpretability task.** Given an EEG recording and the corresponding `zhu-transformer` seizure classification, we want to identify the *sources* on which the model based its inference.

**Problem-specific transformation.** We chose Independent Component Analysis Lee and Lee (1998) (ICA) as our transform of choice. ICA isolates the activity of each individual source to a source-specific channel (Independent Component), assuming statistical independence between the sources. This allows the ICA-domain IG to produce attributions for each individual isolated source, thereby providing insights into our interpretability task (Figure 3).

**Remark 7.** *ICA-IG highlights whether `zhu-transformer` inference is based on known components of epileptic seizure activity or other components irrelevant to the seizure, thus further reinforcing **trust** in the model decision.*

## 5.3 FOUNDATION MODEL TIME SERIES FORECASTING

We use TimesFM Das et al. (2024) time-series foundation model to explain forecasting outputs. We perform zero-shot forecasting, without any fine-tuning, on a time series with exponential trend and seasonal components (Figure 4).

**Remark 8.** *A time-series forecasting model should be equally successful in modeling both the trend and the season to achieve a low-error, long-horizon forecast.*

**Interpretability task.** Given a time-series input and the TimesFM forecast, determine whether the trend or the season is more difficult to model in the long-horizon forecast setting.

**Task-specific transform.** To isolate the relevant *concepts* , we chose Seasonal-Trend decomposition using LOESS (STL) Cleveland et al. (1990) to decompose the input time series into trend and seasonal components.

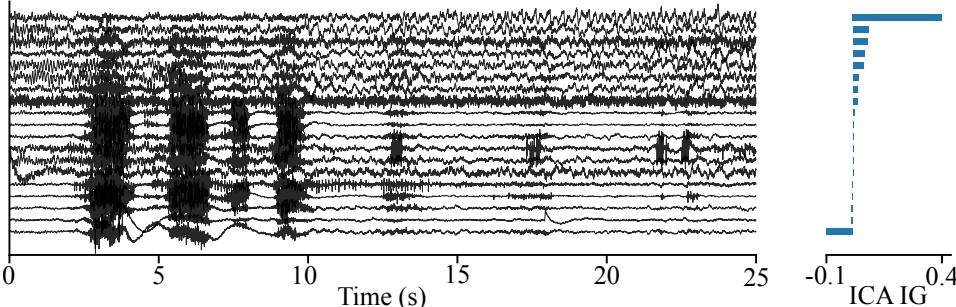

Figure 3: **ICA-domain IG on seizure detection model.** The ICA components are sorted from the component with the highest IG significance (top) to the lowest (bottom). **Left:** 19 output channels calculated from ICA on the original EEG channels. The first channel contains the majority of the epileptic activity, which is visible as an evolving pattern of spike-and-wave discharges at $\sim 4.5$ Hz. Some epileptic activity can also be found in the second channel. Significant muscle artifacts are isolated in the 9th-19th channels between 4 and 10 seconds. **Right:** IG saliency map calculated on the channel components. The map identifies the first channel as the most significant channel in detecting this sample as epileptic. Some significance, although much less, is also given to the next four channels. The channels corresponding to interference components do not get any significance in the output of the classifier. The last channel *tends to tilt* the classifier towards a non-epileptic output.

This attribution domain allows us to study the model's behavior for long-term forecasting horizons where the forecast error increases: the model underestimates the overall trend, while the estimation of the seasonal component presents a smaller error.

**Remark 9.** *Seasonal-Trend IG reveals that TimesFM underweights the trend, degrading long-horizon forecasts. This offers concrete insights to improve model behaviour.*

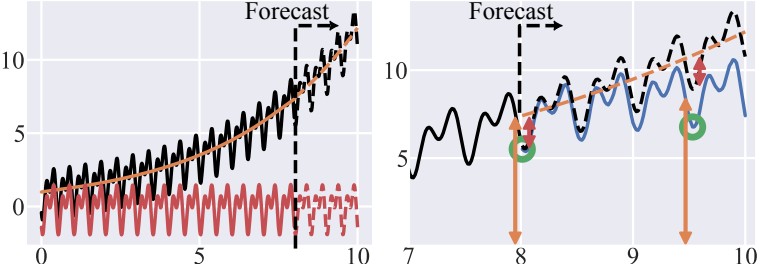

Figure 4: **Seasonal-Trend IG on time series foundation model. Left: Input** time series decomposed via STL into **trend** and **seasonality**. **Right:** Zero-shot **forecasting** using TimesFM with **Seasonal-Trend** IG. For a small horizon, one step ahead prediction (**first circle**), TimesFM forecasts accurately. Of output, 7.5 units are attributed to trend ($\Updownarrow$), aligning with ground truth (dashed orange) and similarly $-1.96$ units to seasonality ($\Updownarrow$). For a longer horizon (**second circle**) the forecast absolute error rises from 0.2 to 2.14. Most of it stems from the model's underestimation of the **trend** (21% relative error), while the **seasonal** effect is correctly captured by the model (5.1% relative error).

## 6    DISCUSSION

Across the three applications presented in Section 5, Cross-Domain IG produces saliency maps that align with the way practitioners typically reason about time-series signals. Feature-level insertion/deletion experiments (Appendix G) support the view that component-domain attributions are both more concentrated and more faithful than time-domain IG and random baselines: removing only a small fraction of the top-ranked components in the chosen domain leads to large changes in the model outputs, whereas adding them back quickly restores the original predictions. We also demonstrate the competitiveness of Cross-Domain IG relative to SoTA methods. On the AudioMNIST benchmark, where we can compare against frequency-domain FreqRISE and IG/LRP-based virtual inspection

layers, Cross-domain IG achieves comparable faithfulness and complexity metrics (Appendix F). This indicates that our method is competitive with existing state-of-the-art saliency methods while remaining applicable to a wider class of complex-valued transforms.

**Limitations.**    While Cross-domain IG addresses the misalignment between time-domain saliency maps and latent structure, it inherits several generic limitations of the IG. Throughout this work we use a straight-line integration path and a single zero baseline; alternative paths or baselines can change the quantitative attributions, and existing variants of IG that stabilise these choices are directly applicable but not explored here. Our framework also assumes an invertible, differentiable transform and is instantiated only with transforms for which this assumption is reasonable, e.g., Fourier, ICA, and seasonal-trend decomposition. Widely used non-invertible or only approximately invertible representations are currently outside our guaranties. Finally, our method presupposes that the practitioner can choose a meaningful explanation domain. If this choice is poor or based on incorrect prior knowledge, Cross Domain IG will still produce clean-looking saliency maps that may be semantically misleading (see Appendix L for a detailed discussion).

## 7    CONCLUSIONS

We introduced a novel generalization of the Integrated Gradients method, which enables saliency map generation in any invertible, differentiable transform domain, including complex spaces. As transforms capture high-level interactions between input points, our methods enhance model explainability, especially in time-series data where individual time-point features are often uninformative. We demonstrated the versatility of Cross-domain Integrated Gradients, applying it to a diverse set of time-series tasks, model architectures, and explanation target domains. Fields where time signals are extensively used, such as healthcare, finance, and environmental monitoring, could benefit from domain-specific saliency maps. In particular, with the recent rise of time-series foundation models, our method provides a powerful investigative tool for examining model behavior. We release an open-source library to enable broader adoption of cross-domain time-series explainability.

## 8    ETHICAL STATEMENT

Risks may arise if the selected explanation target domain is not appropriate or if saliency maps are over-interpreted. It is important to note that the saliency map provides only feature significance scores. Interpreting these scores requires domain expertise. We encourage a holistic interpretative approach to integrating domain knowledge with cross-domain saliency maps. We also caution that this method alone cannot function as definitive proof of the behavior of the model. Responsible usage of the method should take into consideration model, data, and transformation limitations, especially in high-stakes settings such as in healthcare. We elaborate on the limitations of our method in Appendix L

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

## A    CROSS-DOMAIN IG ALGORITHMS

---

**Algorithm 1** Real Target Domain IG

---

**Input:** $f(\cdot), x, \hat{x}, n_{iter}$
**Output:** $IG$
1: $i \leftarrow 1$
2: $sum \leftarrow 0$
3: $tape \leftarrow tensorflow.GradientTape()$
4: $X' \leftarrow T(\hat{x})$
5: **for** $i \leq n_{iter}$ **do**
6:     $z \leftarrow T(x)$
7:     $z \leftarrow \hat{z} + (z - \hat{z}) \cdot i/n_{iter}$
8:     tape.watch($z$)
9:     $x_{rec} \leftarrow T^{-1}(z)$
10:     $y \leftarrow f(x_{rec})$
11:     $dy \leftarrow tape.gradient(y, z)$
12:     $sum \leftarrow sum + dy$
13:     $i \leftarrow i + 1$
14: **end for**
15: $sum \leftarrow sum/n_{iter}$
16: $IG = (z - \hat{z}) \cdot sum$

---

**Algorithm 2** Complex Target Domain IG

---

**Input:** $f(\cdot), x, \hat{x}, n_{iter}$
**Output:** $IG$
1: $i \leftarrow 1$
2: $sum\_real \leftarrow 0$
3: $sum\_imag \leftarrow 0$
4: $tape\_real \leftarrow tensorflow.GradientTape()$
5: $tape\_imag \leftarrow tensorflow.GradientTape()$
6: $\hat{z} \leftarrow T(\hat{x})$
7: **for** $i \leq n_{iter}$ **do**
8:     $X \leftarrow T(x)$
9:     $z \leftarrow \hat{z} + (z - \hat{z}) \cdot i/n_{iter}$
10:     $re\_z \leftarrow \text{Re}\{z\}$
11:     $im\_z \leftarrow \text{Im}\{z\}$
12:     $tape\_real.watch(re\_z)$
13:     $tape\_imag.watch(im\_z)$
14:     $\hat{z} \leftarrow re\_z + j \cdot im\_z$
15:     $x_{rec} \leftarrow T^{-1}(\hat{z})$
16:     $y \leftarrow f(x_{rec})$
17:     $re\_dy \leftarrow tape\_real.gradient(y, re\_z)$          $\triangleright$ Calculate $\frac{\partial g}{\partial p_i}$
18:     $im\_dy \leftarrow tape\_imag.gradient(y, im\_z)$          $\triangleright$ Calculate $\frac{\partial g}{\partial q_i}$
19:     $sum\_real \leftarrow sum\_real + re\_dy$
20:     $sum\_imag \leftarrow sum\_imag + im\_dy$
21:     $i \leftarrow i + 1$
22: **end for**
23: $sum\_real \leftarrow sum\_real/n_{iter}$
24: $sum\_imag \leftarrow sum\_imag/n_{iter}$
25: $IG = \text{Re}\{z - \hat{z}\} \cdot sum\_real + \text{Im}\{z - \hat{z}\} \cdot sum\_imag$

---

---

**Algorithm 3** Complex Target Domain IG with complex differential

---

**Input:** $f(\cdot)$, $x$, $\hat{x}$, $n_{iter}$
**Output:** $IG$
 1: $i \leftarrow 1$
 2: $sum \leftarrow 0$
 3: $tape \leftarrow tensorflow.GradientTape()$
 4: $\hat{z} \leftarrow T(\hat{x})$
 5: **for** $i \leq n_{iter}$ **do**
 6:    $z \leftarrow T(z)$
 7:    $z \leftarrow \hat{z} + (z - \hat{z}) \cdot i/n_{iter}$
 8:    tape.watch($z$)
 9:    $x_{rec} \leftarrow T^{-1}(z)$
10:    $y \leftarrow f(x_{rec})$
11:    $dy \leftarrow tape.gradient(y, X)$
12:    $sum \leftarrow sum + \overline{dy}$
13:    $i \leftarrow i + 1$
14: **end for**
15: $sum \leftarrow sum/n_{iter}$
16: $IG = 2 \operatorname{Re}\{(z - \hat{z}) \cdot sum\}$

---

## B    PROOF OF LEMMA 4.1

**Lemma.** *Let $g : \mathbb{C}^n \to \mathbb{R}$, $\boldsymbol{z} = \boldsymbol{p} + j\boldsymbol{q}$, with $\boldsymbol{p}, \boldsymbol{q} \in \mathbb{R}^N$, $\boldsymbol{\gamma}(t) = \hat{\boldsymbol{z}} + t(\boldsymbol{z} - \hat{\boldsymbol{z}}), t \in [0,1]$ the line from the baseline point $\hat{\boldsymbol{z}}$ to the input point $\boldsymbol{z}$ and $\boldsymbol{n}(t) = \operatorname{Re}\{\boldsymbol{\gamma}(t)\}$ and $\boldsymbol{m}(t) = \operatorname{Im}\{\boldsymbol{\gamma}(t)\}$, $\boldsymbol{n}(t), \boldsymbol{m}(t) \in \mathbb{R}^n$. Then the IG of $g$ in $\boldsymbol{z}$ is given by:*

$$IG_i^{\mathbb{C}^n}(\boldsymbol{z}) = \int_0^1 \left( \frac{\partial g}{\partial p_i} n_i'(t) + \frac{\partial g}{\partial q_i} m_i'(t) \right) dt \tag{8}$$

*Proof.* Let $u : \mathbb{R}^{2n} \to \mathbb{R}$ such that $g(\boldsymbol{z}) = u(\boldsymbol{w}), \forall \boldsymbol{z} = \boldsymbol{p} + j\boldsymbol{q}, \boldsymbol{w} = [\boldsymbol{p}, \boldsymbol{q}]$. For the differential form of $u$:

$$du := \sum_{i=0}^{2N} \frac{\partial u}{\partial w_i} dw_i \tag{9}$$

Similarly to the $g(\boldsymbol{z})$–$u(\boldsymbol{w})$ equivalence, we consider the equivalence between $\boldsymbol{\gamma}(t)$ and $\boldsymbol{a}(t) = [\boldsymbol{n}(t), \boldsymbol{m}(t)] \in \mathbb{R}^{2n}$. Then the pullback of $du$ by $\boldsymbol{a}$ is :

$$\boldsymbol{a}^* du := \sum_{i=0}^{2N} \frac{\partial u}{\partial w_i} a_i'(t) dt \tag{10}$$

Denoting with $a_i'$ the i-th element of $d\boldsymbol{a}/dt$. The line integral of $u$ along the line defined by $\boldsymbol{a}$ is:

$$\int_\gamma du = \int_\gamma \boldsymbol{a}^* du = \int_0^1 \sum_{i=0}^{2N} \frac{\partial u}{\partial w_i} a_i'(t) dt = \sum_{i=0}^{2N} \int_0^1 \frac{\partial u}{\partial w_i} a_i'(t) dt \tag{11}$$

Due to the equivalence between $\boldsymbol{w}$ and $\boldsymbol{p}, \boldsymbol{q}$, and $u$ and $g$, the latter sum can be formulated as :

$$\int_\gamma du = \sum_{i=0}^{N} \left( \int_0^1 \frac{\partial g}{\partial p_i} n_i'(t) dt + \int_0^1 \frac{\partial g}{\partial q_i} m_i'(t) dt \right) = \sum_{i=0}^{N} \int_0^1 \left( \frac{\partial g}{\partial p_i} n_i'(t) + \frac{\partial g}{\partial q_i} m_i'(t) \right) dt \tag{12}$$

, which concludes the derivation.      $\square$

## C    DERIVATION OF DEFINITION 4.1

From Lemma 4.1, we conclude with Definition 4.1 by considering $g(\boldsymbol{z}) = f\left(T^{-1}(\boldsymbol{z})\right)$ and the complex differential form Range (1998):

$$dg = \partial g + \overline{\partial} g \tag{13}$$

with $\partial g = \sum \partial g / \partial z_i dz_i$, $\overline{\partial} g = \sum \partial f / \partial \overline{z_i} \overline{dz_i}$. The complex partial derivatives are defined as Range (1998) $\partial / \partial z_i = 1/2(\partial/\partial p - j\partial/\partial q)$ and $\partial / \partial \overline{z_i} = 1/2(\partial/\partial p + j\partial/\partial q)$. Then the pullback of $dg$ by $\gamma$ is :

$$\gamma^* dg = \sum \frac{\partial g}{\partial z_i} \gamma_i'(t) dt + \sum \frac{\partial g}{\partial \overline{z_i}} \overline{\gamma_i'(t)} dt \tag{14}$$

Since $g \in \mathbb{R}$, $\partial g / \partial \overline{z} = \overline{(\partial g / \partial z)}$; thus:

$$\gamma^* dg = 2 \operatorname{Re}\left\{ \sum \frac{\partial g}{\partial z_i} \gamma_i'(t) dt \right\} \tag{15}$$

Expanding the product into its real and imaginary parts produces the same form as eq. 12:

$$\gamma^* dg = 2 \operatorname{Re}\left\{ \sum \frac{1}{2} \left( \frac{\partial g}{\partial p_i} - j\frac{\partial g}{\partial q_i} \right) (n_i' + jm_i'(t)) \, dt \right\} = \sum \left( \frac{\partial g}{\partial p_i} n_i'(t) + \frac{\partial g}{\partial q_i} m_i'(t) \right) \tag{16}$$

Therefore, the complex integrated gradient definition can be rewritten as:

$$IG_i^{\mathbb{C}^n} = 2 \int_0^1 \operatorname{Re}\left\{ \frac{\partial g}{\partial z_i} \gamma_i'(t) \right\} dt \tag{17}$$

# D RELATIONSHIP BETWEEN FREQUENCY-DOMAIN IG AND FREQUENCY RESPONSE

We probe the two convolutional channels of section 3.2 with sinusoidal signals at varying frequencies, $\xi_i$:

$$x_i(t) = cos(2\pi\xi_i t + \phi) \tag{18}$$

For each input, we perform frequency-domain IG, which yields a saliency map described by eq. 7. We aggregate all produced IGs and compare them to each filter's frequency response:

$$b_i = \| \sum_n w_n e^{-2\pi\xi_i n} \| \tag{19}$$

The results are presented in Figure 5.

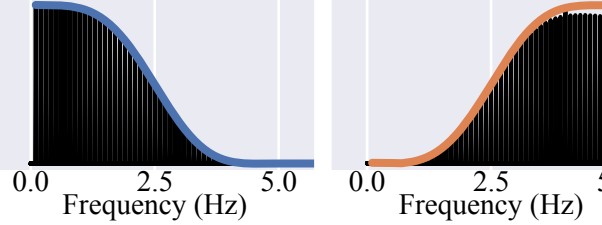

Figure 5: Frequency response (blue - orange) and frequency integrated gradients (black) for the two channels of the model of Section 3.2. We probe the model, performing frequency IG on samples with varying base frequencies.

# E RELATION TO VIRTUAL INSPECTION LAYERS

We demonstrate here the equivalence between Eq. 6 and the Virtual Inspection Layer Vielhaben et al. (2024) for the case of the Discrete Fourier Transform (DFT) domain saliency maps.

Denote the DFT transform $z = Tx$ with :

$$T_{nk}^{-1} = \frac{1}{\sqrt{N}} e^{2\pi kn/N} \tag{20}$$

Thus, from eq.6

$$IG_k^{DFT} = 2\int_0^1 \text{Re}\left\{\sum_{n=0}^{N-1} \frac{\partial f}{\partial x_n} T_{nk}^{-1}(z_k - \hat{z_k})\right\}dt = \sum_{n=0}^{N-1} \text{Re}\{T_{nk}^{-1}(z_k - \hat{z_k})\}2\int_0^1 \frac{\partial f}{\partial x_n}dt$$

$$= 2\sum_{n=0}^{N-1} \text{Re}\{T_{nk}^{-1}(z_k - \hat{z_k})\}\frac{IG_n}{x_n - \hat{x}_n}$$

Denoting $(z_k - \hat{z_k}) = r_k e^{j\phi k}$ then

$$\text{Re}\{T_{nk}^{-1}(z_k - \hat{z_k})\} = \frac{r_n}{\sqrt{N}}cos\left(\frac{2\pi kn}{N} + \phi_k\right) \tag{21}$$

And finally,

$$R_k = 2r_k \sum cos\left(\frac{2\pi kn}{N} + \phi_k\right)\frac{R_n}{x_n - \hat{x}_n} \tag{22}$$

Which is equivalent to the method of Vielhaben et al. (2024).

As an example, we present (Figure 6) the frequency attributions of Figure 2, comparing the Cross-Domain IG in the frequency domain with the Virtual Inspection Layer on top of the time-domain IG.

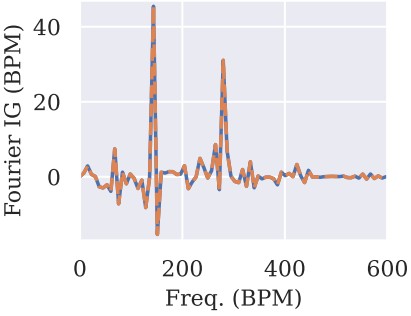

Figure 6: Example of Cross-Domain IG in the frequency domain compared to the Virtual Inspection Layer operating on the time-domain IG.

## F    RELATION TO FREQRISE

We experimentally compared Cross-domain IG (frequency domain) to FreqRISE Brüsch et al. (2024). We run their benchmarking on the AudioMNIST dataset Becker et al. (2024), evaluating Faithfulness and Complexity as defined in Brüsch et al. (2024).

Additionally, we evaluated Cross-domain IG in the Complex Spectrum domain, where the transform $T$ is defined as:

$$T = \mathcal{F}^{-1}\{log(\mathcal{F}\{x(t)\})\} \tag{23}$$

$\mathcal{F}$ is the Fourier transform. The results are presented in Table 1.

## G    FEATURE-LEVEL INSERTION-DELETION

We perform insertion-deletion evaluation tests on the three examples presented in Section 5. Our evaluation indicates that component-level attributions provide more faithful and concentrated evidence for the models' predictions than time-domain attributions: adding top-rated component features rapidly reconstructs the output, while removing them destroys it.

|  | Frequency | | Complex Spectrum | |
|---|:---:|:---:|:---:|:---:|
|  | **Digit** | **Gender** | **Digit** | **Gender** |
| Faithfulness ↓ | | | | |
| **FreqRISE** | 0.160 | 0.416 | - | - |
| **LRP** | 0.205 | 0.431 | - | - |
| **IG** | 0.252 | 0.428 | - | - |
| **IG (our reproduction)** | 0.217 | 0.443 | - | - |
| **CDIG (ours)** | 0.19 | 0.446 | 0.097 | 0.505 |
| Complexity ↓ | | | | |
| **FreqRISE** | 8.17 | 8.01 | - | - |
| **LRP** | 5.84 | 5.16 | - | - |
| **IG** | 6.41 | 4.74 | - | - |
| **IG (our reproduction)** | 6.39 | 4.728 | - | - |
| **CDIG (ours)** | 6.31 | 4.609 | 4.219 | 4.063 |

Table 1: Comparison of Cross-domain IG with FreqRISE. LRP and IG refer to the use of a Virtual Inspection Layer Vielhaben et al. (2024) on top of the time-domain LRP and IG, respectively. The faithfulness and complexity scores for FreqRISE, LRP and IG are taken from Brüsch et al. (2024). We reproduced the IG scores, marked as (our reproduction). Since FreqRISE, LRP and IG are instantiated only in the frequency and time-frequency domains, they cannot provide saliency maps in the Complex Cepstrum domain.

## G.1 HEART RATE EXTRACTION FROM PHYSIOLOGICAL SIGNALS

We follow the procedure outlined below:

1. **Select $k\%$ features**, either in the time or frequency domain. For the frequency and time domain IG, we select the $k$ components with the highest IG score. For the random intervention, we randomly sample $k\%$ unique frequency bins.

2. **Insert/delete** $k$ components to generate modified samples $x_{mod}$.

3. **Infer** heart rate with $x_{mod}$ input.

4. **Compare** $f(x_{mod})$ with the original heart rate inference before any interventions $f(x)$.

An example of inference after inserting/deleting input features is presented in Figure 7. We plot the heart rate inference throughout the entire 2-hour session of subject 15 from the PPG-Dalia dataset. The results for the entire PPGDalia dataset are summarised in Table 2.

| **Top k%-features** | **3.125** % | **25**% | **50**% |
|---|:---:|:---:|:---:|
| Deletion ↑ | | | |
| **Frequency IG** | 66.39 | 133.56 | 127.13 |
| **Time IG** | 10.13 | 50.86 | 104.84 |
| **Random** | 8.53 | 37.03 | 68.34 |
| Insertion ↓ | | | |
| **Frequency IG** | 37.98 | 20.08 | 9.86 |
| **Time IG** | 94.58 | 57.27 | 58.61 |
| **Random** | 123.71 | 100.39 | 66.67 |

Table 2: Insertion-deletion evaluation dropping the k% most important features. Deletion/Insertion distance (expressed in Beats per Minute- BPM) from the original HR inference averaged across 15 subjects of PPGDalia.

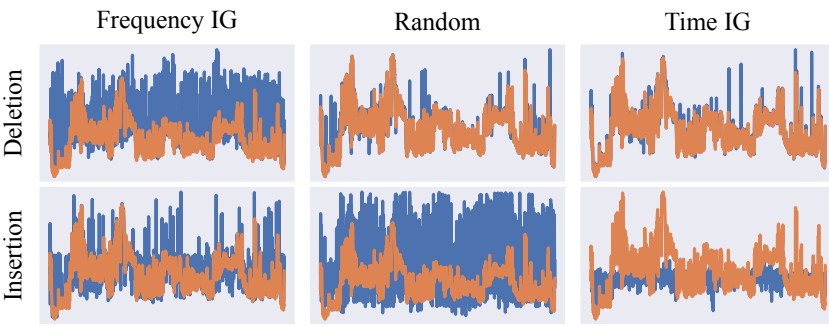

Figure 7: **Example of heart rate inference after deleting features.** We plot the entire session of subject 15 from PPGDalia. For each insertion/deletion, we retain/delete 3.125% of the input features. For the Fourier and time IG these are the frequency bins and time-points with the highest assigned IG score. In the random case, we randomly drop 3.125% of the frequency bins. We plot the **original HR inference** over the duration of the session and the model's output after **modifying the input** accordingly.

G.2    ELECTROENCEPHALOGRAPHY-BASED EPILEPTIC SEIZURE DETECTION

We used the Physionet Siena Scalp EEG Database v1.0.0 Detti (2020); Detti et al. (2020); Goldberger et al. (2000). For each subject's sessions, we retrieved the first sample that is detected as a seizure by the `zhu-transformer`. For each sample, we generated ICA-domain IG saliency maps and performed insertions/deletions with the most important IC. We kept track of the change in the seizure classification probability, $\Delta p = p(\boldsymbol{x}_{mod}) - p(\boldsymbol{x})$, as we:

1. Delete the most important component and perform inference,
2. Maintain the most important component, delete the rest of the components, and perform seizure classification.

We compared these results with those obtained from randomly choosing an IC component and performing the same insertion/deletion evaluation.

|  | ICA IG | Random |
|---|---|---|
| **Deletion** $\Delta p \uparrow$ | 0.1776 | 0.0083 |
| **Insertion** $\Delta p \downarrow$ | 0.0696 | 0.4396 |

Table 3: Insertion-deletion evaluation on the seizure detection model.

## H    EXAMPLE TIME-DOMAIN ATTRIBUTIONS

Figures 8, 9 and 10 present the time-domain attributions from the examples of Section 5. In all three cases, interpreting the time-domain saliency maps is difficult and of limited utility.

**Heart rate inference.** The time-domain IG highlights individual time-points of the PPG input. However, it is difficult to assess:

1. *Does an individual time-point contribute to the heart or interference components?* In the time domain, both the effect of the heart and the interference are mixed, and each time point contains information from both of these components. In contrast, in images, when there is component (object) overlap, one component blocks the other, and a single pixel carries single-component information.

2. *Which time-points should be the most important/influential?* From domain knowledge we know that oscillations around the ground truth heart rate should be the ones affecting the model's output. However, we do not have any such insights in the time domain, and the component overlap further complicates oscillation identification in time.

Consequently, these saliency maps do not allow us to answer the interpretability task of Section 5.1.

**Seizure detection.** Similarly to the heart rate example, it is not easy to visually identify the seizure-related oscillations in the time-domain saliency map.

**Time series forecasting.** The time-domain IG highlights mostly the last input time-points.

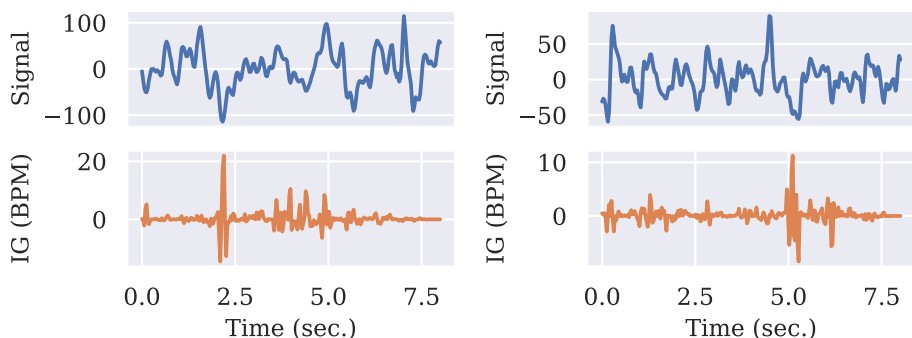

Figure 8: **Time-domain IG for HR inference**. We present the same two inputs as in Figure 2. For each time point in the input we assign a significance value. **Top:** Raw time-domain input which is processed by the model. **Bottom:** IG saliency map expressed in the original time domain.

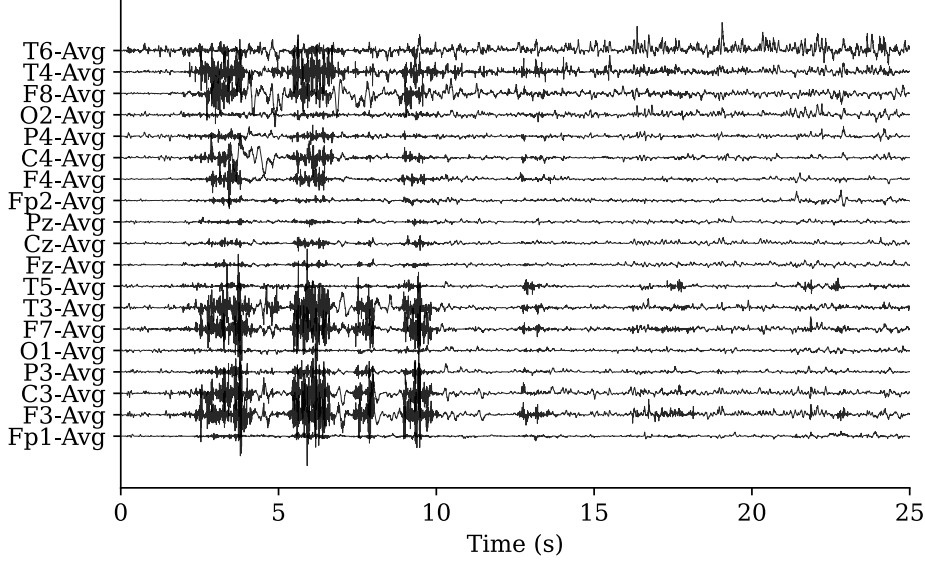

Figure 9: **Time-domain IG for seizure classification**. For each time point on each channel we assign a significance value.

## I  ADDITIONAL EXAMPLES

We present additional Cross-domain IG examples in Figures 11, 12 and 13.

## J  EEG AND ICA

The raw EEG input is presented in Figure 14.

The implementation of the `zhu-transformer` we used can be found here `https://github.com/esl-epfl/zhu_2023`.

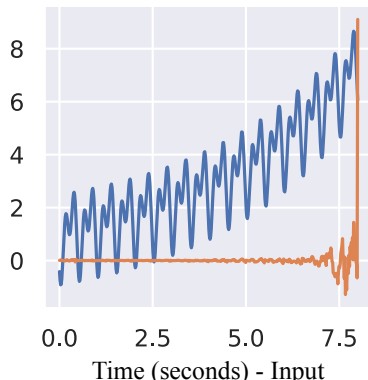

Figure 10: **Time-domain IG for time-series forecasting**. We plot the raw time-domain **input** along with the **IG importance** for each time-point in the input.

The application of ICA in EEG signals is based on the general assumption that the EEG data matrix $X \in \mathbb{R}^{N \times M}$ is a linear mixture of different sources (activities) $S \in \mathbb{R}^{N \times M}$ with a mixing matrix $A \in \mathbb{R}^{N \times N}$ such that $X = AS$, where $N$ is both the number of sources and EEG channels, and $M$ is the number of samples in the dataset. Sources are assumed to be statistically independent and stationary. These assumptions can be leveraged to compute an inverse unmixing matrix $W = A^{-1} (\in \mathbb{R}^{N \times N})$, such that $S = WX$. Finding $W$ is an ill-posed problem without an analytical solution, which can be estimated by means of different ICA algorithms Hyvärinen et al. (2001); Klug and Gramann (2021). ICA is used in EEG to decompose the signal into independent components that separate the signal of interest from various sources of artifacts Winkler et al. (2011). In this work, for ICA we selected the FastICA algorithm implemented in `sklearn` (`max_iter` $= 3 \cdot 10^4$, `tol` $= 1 \cdot 10^{-8}$).

The independent channels estimated using ICA are presented in Figure 15.

## K  GENERATED TIME SERIES FOR TIMESFM FORECASTING

We generate a synthetic time series signal, $x(t)$, composed of an exponential trend, $x_{trend}(t)$, and a seasonal component, $x_{seasonal}(t)$:

$$x_{trend}(t) = e^{\frac{t}{\alpha}}$$
$$x_{seasonal}(t) = sin(2\pi \cdot \xi \cdot t + \phi) + sin(2\pi \cdot 2\xi \cdot t + \phi)$$
$$x(t) = x_{trend}(t) + x_{seasonal}(t)$$

For the example in Section 5.3 $\alpha = 4$, $\xi = 2Hz$. For the samples presented in Appendix I they were randomly sampled from $\alpha \sim U(4.0, 7.0)$ and $\xi \sim U(3.0, 8.0)[Hz]$. A window of 512 time points, starting at $t = 0$, are given as input to TimesFM which generates forecasts up to 128 time points in the future from $t = 512$. The input time series and STL decomposition are presented in more detail in Figure 16.

## L  LIMITATIONS

In this work, we have addressed the limitations of IG regarding time-domain saliency maps. The rest of the original IG limitations are also transferred to our method. For example, the current implementation focuses on a linear integration path, reflecting the original IG. However, other non-linear paths, e.g., Guided IG Kapishnikov et al. (2021), should be explored. In our Remarks in Section 4 we briefly note how already available solutions to these limitations could be transferred directly to Cross-domain IG. For clarity, we summarise them here:

1. **Integration path.** In this work, we used a linear path in line with the original IG Sundararajan et al. (2017). However, eq. 6 allows for the use of non-linear curves such as in Yang et al. (2023); Kapishnikov et al. (2021).

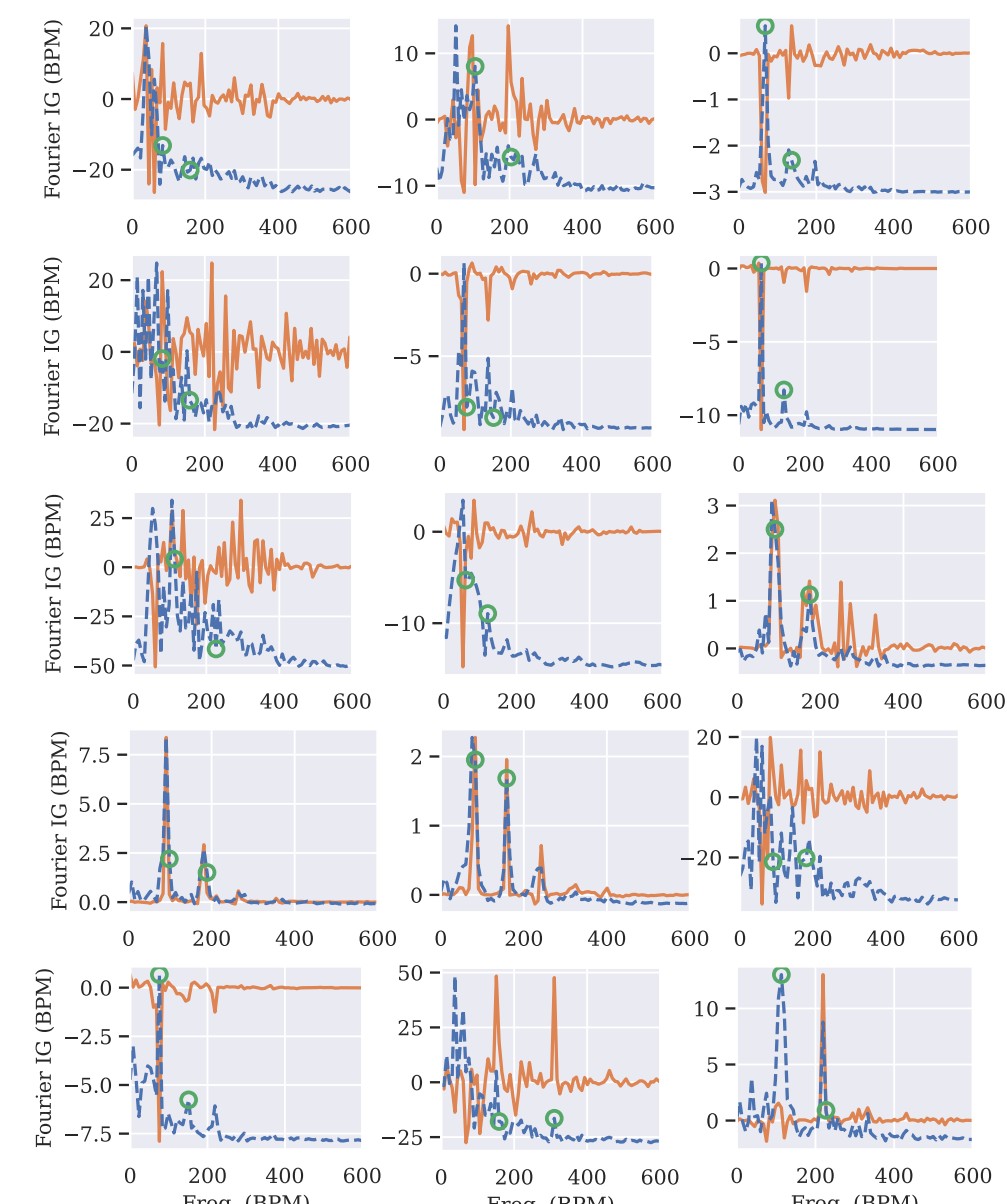

Figure 11: **Frequency-domain IG for heart rate inference model.**

2. **Choosing the baseline.** In Sundararajan et al. (2017) the authors argue that a baseline point exists for most deep networks. In cross-domain IG, if such a point exists, then it can be trivially defined in the target domain through the transform $T$.

3. **Computational overhead.** Similarly to the original IG, our method requires multiple differentiations to approximate the integral (Definition 4.1). We require an additional step due to the transform $T$: computing the inverse and the backpropagation over it.

Our method requires an invertible, differentiable transform and a carefully selected baseline point. Consequently, we excluded non-invertible transforms, and further investigation is needed for approximately invertible cases. Baseline selection also plays a role in the final saliency map. We focused on the zero-signal as the baseline point; future work should include an extensive investigation into the effects of the baseline selection. Finally, multiple transforms can be combined to provide a multi-faceted saliency map, such as ICA combined with frequency domains, and automatic transform

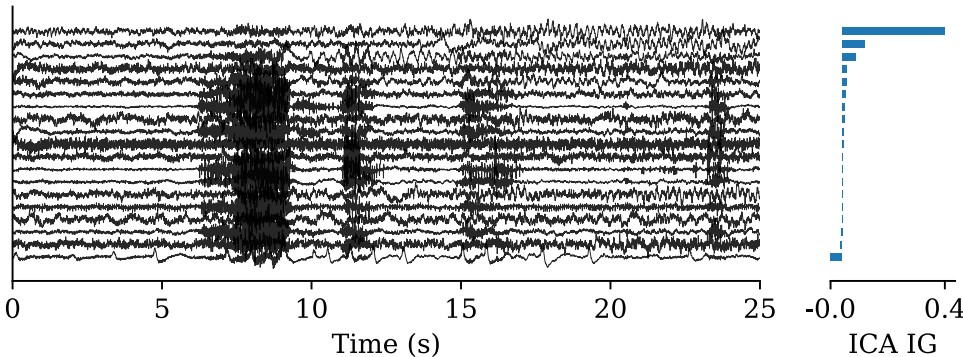

Figure 12: **ICA-domain IG for seizure detection model.** Similarly to the example presented in Section 5.2, the first channel contains the majority of the seizure components. IC channels that contain mostly interference are assigned a very small IG score.

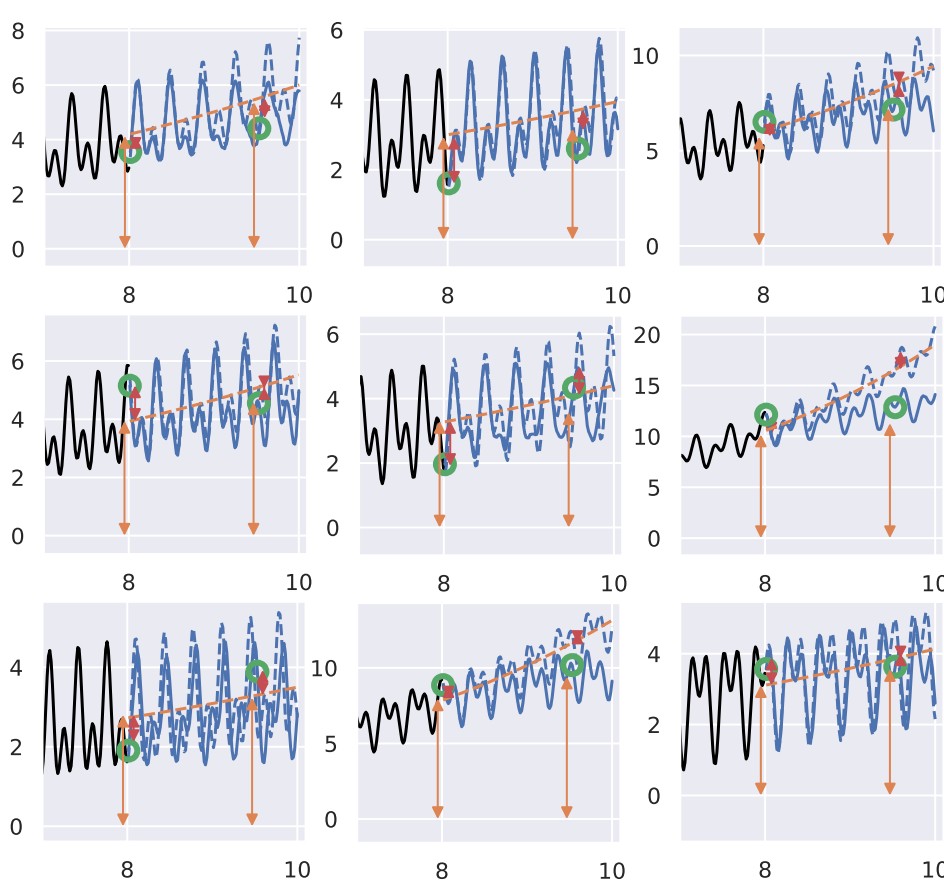

Figure 13: **Seasonal-Trend IG for TimesFM forecasts**. We generate synthetic samples by sampling them as described in Appendix K.

selection could help streamline the process. We leave *ensemble* domains and automatic domain selection as future work.

## M    EXPERIMENTS COMPUTE RESOURCES

All experiments were run on an NVIDIA Tesla V100 with 32 GB of memory.

## N    USE OF LLMS

We used a large language model (LLM) solely for light copy-editing (grammar and wording).

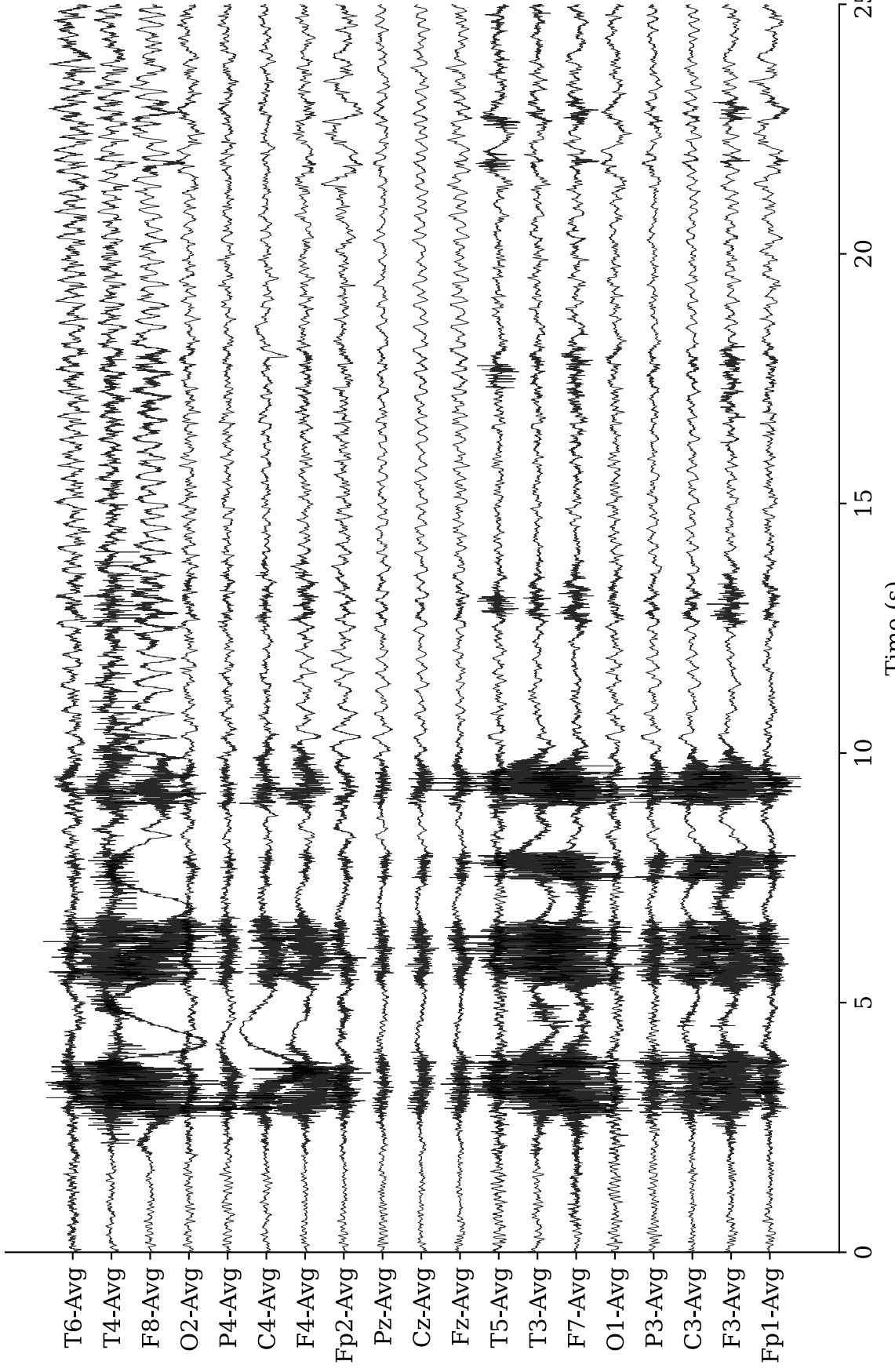

Figure 14: EEG signal in the original channel space.

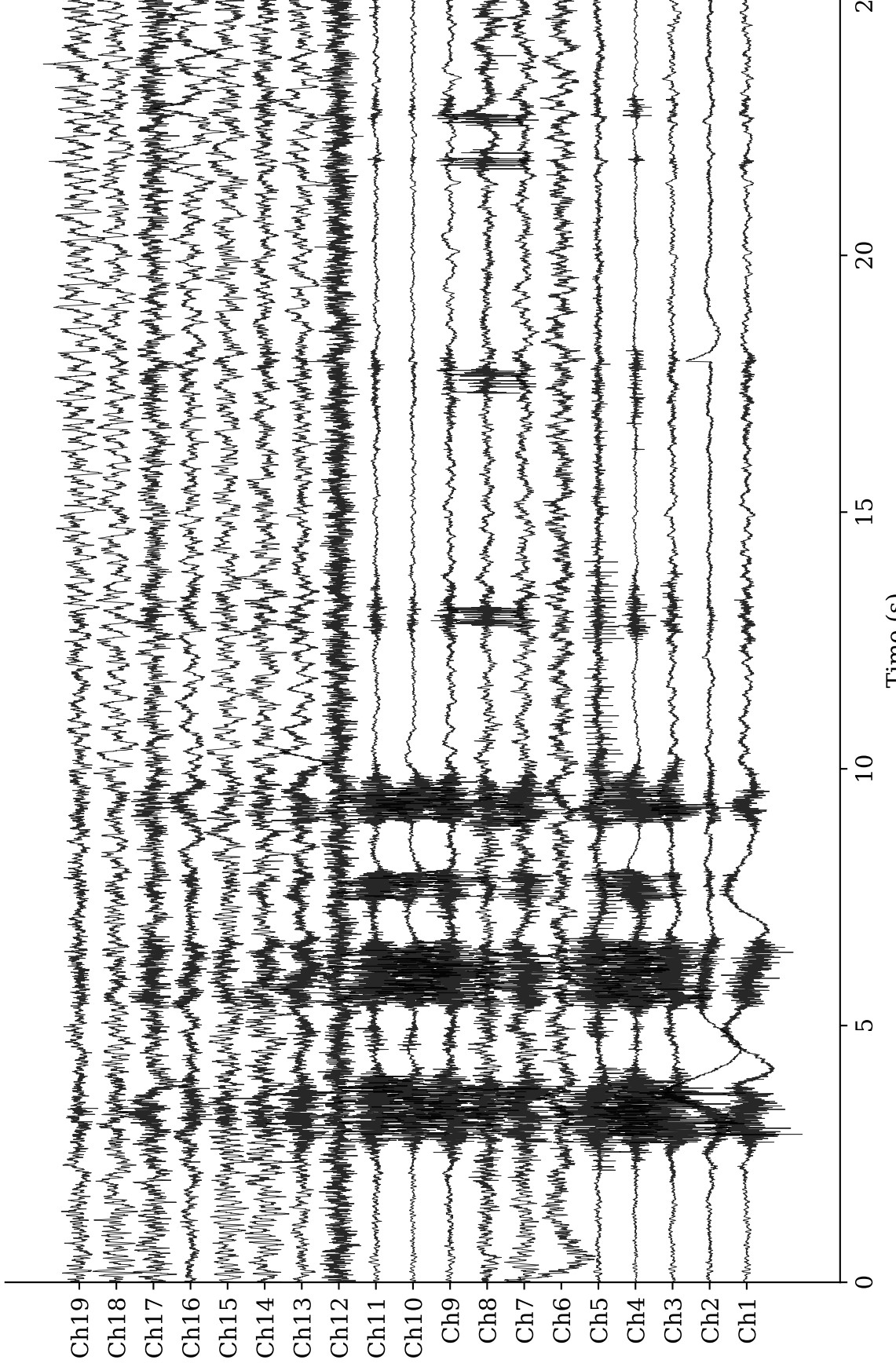

Figure 15: EEG signal in the Independent Component space.

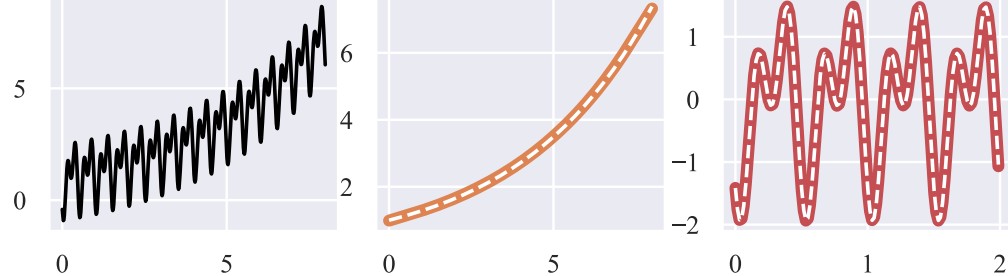

Figure 16: **Input time series for forecasting and successful STL decomposition. Left:** time series with a **trend** and a **seasonal** component. **Center:** The decomposed **trend** component and ground truth trend (white dashed line). **Right:** The decomposed **seasonal** component and ground truth seasonality (white dashed line).

