# OpenReview forum: "Time series saliency maps: Explaining models across multiple domains"
_ICLR.cc/2026/Conference — Submitted to ICLR 2026_

### Official Review · Reviewer_ud4m · 2025-10-23

**Soundness:** 2
**Presentation:** 3
**Contribution:** 1
**Rating:** 2
**Confidence:** 4

**Summary:**

This work is focused on the explainability of black-box model in the context of time series models. The key insight is that the semantically meaningful information might not always be found in the time domain, but in other domains such as the frequency domain. To address this limitation, a generalization of the well-known Integrated Gradients method is proposed such that explanation can be presented in different domains. The proposed methodology is analyzed and evaluated on 3 time series analysis tasks.

**Strengths:**

1. A clear idea that is well motivated.

2. A through theoretical analysis of the proposed methodology.

3. A nicely written and well-structured manuscript.

**Weaknesses:**

1. The novelty is low. The idea of providing explanations in a a different domain is already established. The paper mentions the Virtual Inspection Layers of Vielhaben et al. [1], but do not include it as a baseline, even though [1] also evaluated Integrated Gradients with their Virtual Inspection Layers. Furthermore, several other works [2, 3] have presented methodology for providing explanations in other domains than the time domain.

2. The experimental evaluation is limited. The proposed methodology is tested on 3 datasets, but no baselines are provided, and a limited quantitative evaluation. Compared to existing works [1, 3], where numerous datasets are used and a wide range of explainability metrics are evaluated, the evaluation in this work does not give insights into the usefulness of the proposed method.

- [1] Vielhaben et al., Explainable AI for time series via Virtual Inspection Layers, Pattern Recognition, 2024
- [2] Brüsch et al., FreqRISE: Explaining time series using frequency masking, NLDL, 2025
- [3] Brüsch et al., FLEXtime: Filterbank Learning to Explain Time Series, Explainable Artificial Intelligence, 2025

**Questions:**

1. How does the proposed method quantitatively compare to [1, 2, 3] in terms of established explainablity metrics like faithfulness, localization, complexity, and robustness?

2. Apart from being specific for Integrated Gradients, how is the invertible transform introduced here to transfer between domains different from the transform introduced in [2]?

- [1] Vielhaben et al., Explainable AI for time series via Virtual Inspection Layers, Pattern Recognition 2024
- [2] Brüsch et al., FreqRISE: Explaining time series using frequency masking, NLDL, 2025
- [3] Brüsch et al., FLEXtime: Filterbank Learning to Explain Time Series, Explainable Artificial Intelligence , 2025??

---

> ### Author Response · Authors · 2025-11-23
>
> We thank the reviewer for their thoughtful feedback and for identifying related works. We appreciate the recognition of our theoretical analysis and the structured presentation of our manuscript.
>
> **Response to Novelty Concerns**
>
> The high-level concept of explaining time-series in alternative domains (e.g., frequency) has been explored in prior work, including in the contributions of Vielhaben et al. [1] and Brüsch et al. [2, 3]. However, our contribution is not merely the application of explanation methods to these domains, but the derivation of a unified, axiomatic framework (Cross-Domain Integrated Gradients - CDIG) that generalizes this concept.
>
> Our novelty is distinct from the cited works in four key ways:
>
> - **Axiomatic Unification:** VIL [1]  derives LRP-inspired rules to translate time-domain attributions (from LRP or IG) to the frequency domain - crucially their derivation does not prove IG’s axioms. We derive a general formulation that guarantees the axioms of Completeness and Implementation Invariance for any differentiable, invertible transform—including complex-valued ones. We prove in Appendix E that VIL on IG time-domain outputs is a special case of our method when applied to the Discrete Fourier Transform (DFT). Our framework unifies these approaches analytically.
> - **Gradient-based vs. Perturbation-based:** Works [2] (FreqRISE) and [3] (FLEXtime) are primarily perturbation/masking-based methods. In contrast, CDIG is a purely gradient-based method. These two families of saliency methods approach the attribution problem with different desiderata. IG is defined to fulfill Sensitivity, Implementation invariance and Completeness [8]. In contrast, perturbation-based approaches target Faithfulness and Localisation [9].
> - **Computational efficiency.** Cross-domain IG requires less iterations than FreqRISE or FlexTIME. Specifically, in our experiments we used 300 iterations for Section 5 and 20 iterations for the AudioMNIST benchmark (Appendix F). In contrast, FreqRISE required 10.000 iterations for sampling the masks in the AudioMNIST task [4]. FLEXtime used 1.000 iterations [6]. In the original IG paper [8] the authors mention that 20-300 iterations are enough for IG.
> - **Domain Agnosticism:** The cited works focus predominantly on Frequency and Time-Frequency domains. We demonstrate that CDIG works out-of-the-box on qualitatively distinct domains such as ICA (for EEG), STL Decomposition (for forecasting), and Complex Cepstrum (for audio), using the exact same theoretical machinery.
>
> We clarify this position in the related-work section in the original manuscript. We have updated it to explicitly discuss the relation to FreqRISE [4] and FLEXtime [5].
>
> **On experimental evaluation and baselines**
>
> Our evaluation philosophy follows the original IG paper: we emphasise axiomatic soundness and mechanistic interpretability, and use experiments to support the theory rather than to define it. What we already do:
> - **Theoretical guarantees.** We prove CDIG’s Completeness and path independence, including the extension to complex domains, closely mirroring the validation strategy of IG.
> - **Mechanistic alignment.** On a simple CNN, we derive a closed-form expression showing that “complexIG” in the frequency domain exactly tracks the filters’ frequency responses, providing a tight link between attributions and the model’s internal mechanism.
> - **Three real-world applications.** We apply CDIG to heart-rate estimation from PPG, EEG-based seizure detection, and zero-shot time-series forecasting, each time selecting a domain motivated by signal-processing knowledge. These examples show that CDIG yields semantically meaningful attributions that are not visible in time-domain iG.
> - **Quantitative faithfulness via insertion/deletion.** In Appendix F we perform feature-level insertion/deletion experiments on all three tasks, comparing CDIG to time-domain and random baselines. CDIG attributions are more concentrated and more faithful in the sense that retaining a small fraction of the most important components preserves the model output.
>
> **Regarding direct comparison to [4, 5, 6].**
>
> For VIL + IG [5, 4], our analytic equivalence for the DFT case implies that all faithfulness, localisation, complexity and robustness metrics match CDIG for identical settings, up to numerical approximation.
>
> To address the reviewer’s request for stronger empirical evidence, we have added a new experimental section using the AudioMNIST dataset.  We now compare CDIG against FreqRISE [4], LRP, and standard Time-IG. We evaluate Faithfulness using Insertion/Deletion curves. We found CDIG achieves faithfulness scores comparable to the state-of-the-art perturbation method (FreqRISE). Furthermore, we extend the analysis in [4] to include multiple domains. This is now the first quantitative benchmark for saliency maps in the Complex Cepstrum domain, demonstrating the method's unique versatility.

---

> > ### Author Response · Authors · 2025-11-23
> >
> > **Answers to questions.**
> >
> > **Q1.** For VIL + IG [5, 4], as we prove in the Appendix E, CDIG combined with the DFT is analytically equivalent to IG + VIL. Therefore, theoretically, their scores for faithfulness, localization, and robustness are identical (up to numerical precision differences in implementation). We show this both theoretically and now empirically in the new analysis which includes numerical and graphical comparison.
> >
> > For FreqRISE [4] we have reproduced the experiments of the paper, adding CDIG (DFT) and Complex Cepstrum.
> >
> > **Q2.** The transform itself is the same general idea that is used also in already existing literature [4-6]. What is novel is the way we utilise this transform to generate the saliency map. Here we describe in more detail these differences:
> > In VIL, they derive the translation from the time-domain to the frequency domain based on ideas from Layer-wise Relevance Propagation (LRP). Although VIL can be used on top of time-domain IG, their derivation does not necessarily guarantee that the IG axioms are fulfilled. This is because they based their derivation on LRP, not on IG’s axioms. In our analysis, we show that ultimately CDIG equipped with DFT and IG + VIL are essentially the same. However, the path we have taken is substantially different: we start from the IG’s axioms and  generalise them for complex-domain inputs.
> > In FreqRISE and FlexTIME, the transform is used to define a mask in the transform’s domain.
> >
> > Crucially, in all three works the transform is set to either the frequency or the time-frequency domain. In our paper the situation is different: we treat the transform as a generic mapping with the only constraints being that it should be invertible and differentiable. Our derivations Section 4, Appendices B, C, guarantee that the axiomatic desiderata of IG are fulfilled for any transform that satisfies these requirements.
> >
> > [1] Explainable AI for time series classification: a review, taxonomy and research directions
> >
> > [2] Post-hoc saliency methods fail to capture latent feature importance in time series data
> >
> > [3] What about the latent space? The need for latent feature saliency detection in deep time series classification
> >
> > [4] FreqRISE: Explaining time series using frequency masking
> >
> > [5] Explainable AI for time series via virtual inspection layers
> >
> > [6] FLEXtime: Filterbank Learning to Explain Time Series
> >
> > [7] Audiomnist: Exploring explainable artificial intelligence for audio analysis on a simple benchmark
> >
> > [8] Axiomatic Attribution for Deep Networks
> >
> > [9] Rise: Randomized input sampling for explanation of black-boc models

---

> ### Comment · Reviewer_ud4m · 2025-11-27
> **Response from the reviewer**
>
> I would like to thank the reviewers for a nice and thorough rebuttal, that is clearly formatted and nicely presented. It is clear that a great effort has been put into the rebuttal, and I think it has improved some aspects of the paper.
>
> Nevertheless, most of my concerns still remain. As explained in the rebuttal, CDIG combined with the DFT is analytically equivalent to IG + VIL, and I think extending VIL or other approaches to other domains is a simple extension. I therefore do not see a great benefit in terms of the applications that CDIG opens up. And in terms of the experimental evaluation, there does not seem to be a great benefit in terms of the performance. Then, the contribution of the paper becomes generalising IG to complex valued functions. While this contribution has merit, it is unclear how significant this contribution is.
>
> I again thank the authors for a nice rebuttal, and I think there are aspects of the paper that are noteworthy and valuable for the community. However, I do not believe it has the necessary quality for ICLR. I will therefore keep my original score.

---

> > ### Author Response · Authors · 2025-12-03
> >
> > We thank the reviewer for acknowledging the effort put into the rebuttal and the improvements in the paper. We appreciate the opportunity to clarify the contributions and practical benefits of our work.
> >
> > While we understand the reviewer’s perspective that CDIG and VIL share mathematical similarities in the specific case of the DFT, we argue that viewing CDIG as a “simple extension” of VIL overlooks the fundamental difference in their construction and generalisability. Below, we outline why this distinction matters for practitioners and theoreticians.
> >
> > In the paper we demonstrate that CDIG reduces to IG + VIL for the DFT. We view this as a theoretical validation and not a redundancy. VIL is derived specifically for linear transforms using LRP principles [1, 2]. To apply VIL to a new domain, one must manually derive and justify new relevance propagation rules for that specific transform’s operations. This has been done in the LRP literature for several non-standard layers [4, 5]. However, it is not an automatic process and needs to be explicitly derived and validated for each new transform. In contrast, our CDIG framework is domain-agnostic by definition. We prove that for any invertible, differentiable transform $T$ (linear or non-linear, real or complex), applying IG in the transformed space preserves the fundamental axioms of Completeness and Implementation Invariance. This is a key contribution of our paper as we do not “extend VIL”; we unify cross-domain attribution under a single, rigorous IG definition. The fact that it recovers the VIL formula for the DFT confirms our theory is sound, but our theory applies broadly where VIL does not exist.
> >
> > The reviewer suggests that extending VIL to other domains is “simple”. We respectfully point out that this is not the case for non-linear complex-valued transforms, such as the Complex Cepstrum used in our experiments. The Complex Cepstrum involves non-linear operations (e.g., complex logarithms). To the best of our knowledge, no VIL or LRP rules currently exist for complex-valued non-linearities. Deriving them would require significant theoretical effort to ensure conservation properties are met. In contrast, CDIG handles complex transforms, such as the Complex Cepstrum via the chain rule and our Complex IG formulation, without requiring any new hand-crafted attribution rules. This demonstrates that CDIG is not just a trivial extension, but a generalised solution for complex non-linear domains.
> >
> > The reviewer questioned the significance of our Complex IG contribution. We argue this is a fundamental necessity for modern signal processing as it ensures correct gradient calculus. Our derivation uses Wirtinger calculus to ensure gradients are mathematically consistent in the complex plane. It also offers Axiomatic Guarantees. By strictly generalising IG [6], we ensure that the axioms (e.g. Sensitivity, Implementation Invariance) hold in the complex domain. Ad-hoc extensions of real-values methods do not provide these guarantees. VIL [1] is not guaranteed to preserve IG’s axioms and is not invariant under reparameterisations, such as switching from real/imaginary coordinates to magnitude/phase coordinates.
> >
> > Finally, the reviewer noted a lack of benefit in experimental performance. Our new results on AudioMNIST (added in the rebuttal) provide concrete evidence to the contrary.
> > - **Higher Faithfulness.** Cross-domain IG (using Complex Cepstrum) outperforms both VIL (Frequency) and FreqRISE (Frequency) [3] in faithfulness metrics for digit classification. This proves that flexibility in domain selection directly translates to better explanations.
> > - **Computational Efficiency.** Compared to FreqRISE [3] (a leading perturbation baseline), CDIG is orders of magnitude more efficient. We achieve convergence in 20 iterations, whereas FreqRISE requires ~10.000 masking steps.
> >
> > In summary, our contribution is the establishment of the first unified, axiomatic framework for Cross-Domain IG. It allows practitioners to define any differentiable domain - even non-linear, complex ones like the Complex Cepstrum - and immediately generate axiomatically sound explanations without deriving new propagation rules. This provides a capability and rigorous foundation that VIL and standard IG simply do not offer.
> >
> > [1] Explainable AI for time series via virtual inspection layers
> >
> > [2] On pixel-wise explanations for non-linear classifier decisions by layer-wise relevance propagation
> >
> > [3] FreqRISE: Explaining time series using frequency masking
> >
> > [4] Layer-wise relevance propagation for neural networks with local renormalisation layers
> >
> > [5] Attnlrp: attention-aware layer-wise relevance propagation for transformers
> >
> > [6] Axiomatic Attribution for Deep Networks

---

### Official Review · Reviewer_ywmS · 2025-10-30

**Soundness:** 4
**Presentation:** 4
**Contribution:** 4
**Rating:** 8
**Confidence:** 3

**Summary:**

This paper proposes the Cross domain Integrated Gradients method, which extends the Integrated Gradients (IG) to any reversible and differentiable transformation domain (including the complex domain), providing a more semantic and insightful explanation for time series models.

**Strengths:**

1. Wide universality: The attribution framework for unknown transformations has high universality and is not specific to any particular transformation.

2. Solid theoretical contribution: Extend the IG method to the complex field.

3. Compelling & Diverse Applications: This method shows significant application potential in multiple scenarios like medical and other general time series application.

**Weaknesses:**

Overall, I think the strengths of this paper are very prominent, and the disadvantages are not worth mentioning compared to it. Here are a few of my small concerns.

1. The main text of the paper lacks quantitative experimental comparisons, and the structure should be adjusted by moving the section in Appendix F to the main text.

2. Appendix D indicates that this method is the general form of [1]. Can the author compare the two in a visual form to see if the actual effect is consistent with the theory?

[1] Johanna Vielhaben, Sebastian Lapuschkin, Grégoire Montavon, and Wojciech Samek. Explainable ai
for time series via virtual inspection layers. Pattern Recognition, 150:110309, 2024.

**Questions:**

1. Since this method can be applied to all reversible transformations, is it suitable for the current popular flow generation models? What would be the computational burden in practical applications?

2. What are the errors of this method for differentiable irreversible transformations? Is it possible to make corrections?

---

> ### Author Response · Authors · 2025-11-23
>
> We appreciate the reviewer’s positive assessment and for recognizing the universality and theoretical solidity of our work. We appreciate the thoughtful suggestions to further strengthen the manuscript.
>
> **Moving quantitative experiments (Appendix F) to the main text.**
>
> We agree with the reviewer that the quantitative results are crucial and deserve prominence. In line with the comments of the other reviewers we have strengthened them to include additional analysis on AudioMNIST. In the original submission, we were constrained by the strict 9-page limit, which necessitated moving the insertion/deletion experiments to the Appendix to prioritize the method derivation and diverse qualitative applications. If accepted, we could utilize the additional page allowed in the camera-ready version to move the content of the quantitative comparisons into the main experimental section, as suggested.
>
> **Visual comparison with Virtual Inspection Layers (VIL) [2].**
>
> We thank the reviewer for the suggestion. Since our theory predicts that Cross-Domain IG reduces to the exact same formulation as IG+VIL when applied to the Discrete Fourier Transform, the visual outputs should be identical (within numerical precision). We have added a side-by-side plot in Appendix E of the revised manuscript comparing the saliency maps produced by IG+VIL and CDIG on the DFT domain. As expected, the maps are visually indistinguishable, confirming the theoretical alignment. To further bolster this with quantitative data, we have also added a comparison table including results from FreqRISE [1] as an additional comparison.
>
> **Answers to Questions**
>
> **Q1.** Yes, our method is highly suitable for Flow-based models (Normalizing Flows). Since Flow models are constructed as a sequence of invertible, differentiable transformations with a tractable Jacobian, they satisfy the exact requirements of Cross-Domain IG by design.
>
> Application: One could use CDIG to explain the model's output in the latent space of the flow (using the Flow itself as the transform $T$) or to explain a downstream task using the flow, as long as it acts as semantically meaningful feature extractor.
> Computational Burden: The cost is dominated by the calculation of the gradients through the inverse transform $T^{-1}$. For flow models, the inverse is explicit, but deep. The computational complexity would be roughly proportional to $N_{steps} \times C_{backward}$, where $N_{steps}$ is the number of integral approximation steps (usually 50-100) and $C_{backward}$ is the cost of a standard backward pass through the flow. This is computationally feasible and comparable to running standard IG on a deep network.
>
> **Q2.** The core requirement of CDIG is the existence of a valid inverse map $x = T^{-1}(z)$ to define the path integration $\gamma(t)$ in the input space corresponding to the path in the latent space. If $T$ is non-invertible (e.g., dimension reduction), the inverse problem is ill-posed (one $z$ maps to multiple $x$'s). To address this, one would have to define a specific "pseudo-inverse" or a generative mapping (e.g., a decoder $D(z) \approx x$). The "error" introduced would be the reconstruction error $||x - D(T(x))||$. If the transformation loses critical information required by the model for prediction, the saliency map in the domain $Z$ might fail to capture the cause of the prediction.
>
> [1] FreqRISE: Explaining time series using frequency masking
>
> [2] Explainable AI for time series via virtual inspection layers

---

> > ### Comment · Reviewer_ywmS · 2025-11-24
> >
> > Thanks for the author's response. Although I have a very positive attitude towards this work, I also agree with the concerns of other reviewers about the lack of performance comparison with existing methods. At the same time, I also noticed that in the comparison in Appendix F, the author found that this method does not have a significant advantage over other methods. However, I believe that developing a universal multi domain explaination method (including theoretical exploration) is still an important contribution in XAI community, so I will maintain my score.
> >
> > Minor issues: regarding the response to Q2, I believe that training a decoder may be difficult to achieve ideal results because you do not have appropriate signals to guide the training. However, defining pseudo inverse separately seems to undermine the universality of the method. This may require further exploration by the XAI communities.

---

### Official Review · Reviewer_YxUK · 2025-11-01

**Soundness:** 2
**Presentation:** 2
**Contribution:** 2
**Rating:** 2
**Confidence:** 4

**Summary:**

The paper provides a novel explainability method for the time series domain. Specifically, the method is based on an extension of the popular saliency method Intergrated Gradients to incorporate multiple domains that can be derived through an invertible, differential transformation from the time domain. Importantly, the proposed method maintains the sensitivity and implementation invariance properties of IG.

**Strengths:**

- The paper proposes a saliency method for time series which does not solely focus on the time domain, but can also integrate latent features such as frequencies
- The presentation of the method is easy to follow
- The paper provides open access to the method in the form of a Python package

**Weaknesses:**

- The paper completely lacks references in the introduction. This is not in line with good research practice. It is unclear to the reader whether observations and statements are taken from the literature or are a novel contribution by the paper. Importantly, the observation that existing saliency maps fail in the
time-series domain and that other features, e.g., stemming from the frequency domain, are not novel and have been shown before in the literature (e.g., [1],[2],[3]). This renders Proposition 1, without proper citations, almost plagiarism. Section 3.2 is therefore unnecessary. I am not raising an ethics flag at the moment, but this aspect is, in my opinion, sufficient for rejecting the paper without further evaluation.
- The paper does not discuss the many restrictions and limitations of IG and their equivalent part in the proposed method.
- The paper states that the method is applicable to many domains. However, all argumentation and derivation (e.g., section 4.2) solely focus on the frequency domain. More explanation and examples are needed here to evaluate the usefulness of the method.
- The method requires domain knowledge to specify the domain of interest. In practice, such knowledge might not exist, or if it does, might be limited. This can lead to dangerous misinterpretations of the explanations and wrong decision-making. It would be desirable that a novel explainability method can directly infer saliency across many (unspecified) domains to potentially uncover so far unknown important features, instead of suffering similar limitations to existing time series saliency methods on the time domain.
- The paper does not discuss the experimental results or the limitations of the proposed method. Overall, the paper seems unfinished.
- The experimental section only focuses on three examples with specific ML methods. Here, a model-agnostic evaluation would be beneficial.



[1] Schröder, Maresa, Alireza Zamanian, and Narges Ahmidi. "Post-hoc saliency methods fail to capture latent feature importance in time series data." International Workshop on Trustworthy Machine Learning for Healthcare. Cham: Springer Nature Switzerland, 2023.

[2] Schröder, Maresa, Alireza Zamanian, and Narges Ahmidi. "What about the Latent Space? The Need for Latent Feature Saliency Detection in Deep Time Series Classification." Machine Learning and Knowledge Extraction 5.2 (2023): 539-559.

[3] Theissler, Andreas, et al. "Explainable AI for time series classification: a review, taxonomy and research directions." IEEE Access 10 (2022): 100700-100724.

**Questions:**

- Section 4.2: How is the known failure mode of IG addressed in other domains besides the frequency domain?
- How are failure modes/limitations of IG addressed in the proposed method for general ML models (not only CNNs)?
- How can the method integrate multiple domains at the same time? Importantly, how can it detect + explain interactions between the domains?

---

> ### Author Response · Authors · 2025-11-23
>
> We sincerely thank the reviewer for their rigorous assessment of our work. We have taken the concerns regarding methodological attribution and the positioning of our contributions with the utmost seriousness. We agree that clear attribution is the bedrock of scientific integrity.
>
> To address this, we have significantly revised the Introduction, Proposition 1, and Section 3.2 to explicitly ground our work in prior literature we had already presented in the related works and in the additional literature all reviewers kindly provided. We believe these changes resolve the ambiguity regarding the novelty of our specific contribution (the Cross-Domain IG framework) versus established observations (the failure of time-domain saliency).
> Below, we address the specific weaknesses and questions in detail.
>
> **Attribution, Introduction, and Proposition 1**
>
> We respectfully clarify that we did not intend to present the general observation—that time-domain saliency fails to capture latent features—as a novel finding of this paper. This phenomenon was discussed in our original Related Works, where we cited [1]. In Related Works we also cited [5, 7] which acknowledge the problem and address it in the frequency and time-frequency domains.
> However, we acknowledge that by not citing these works in the Introduction and Proposition 1, we inadvertently created ambiguity regarding the origin of this observation. We have rectified this in the revised manuscript:
> - **Introduction**: We now explicitly state that the limitation of time-domain saliency in the presence of latent features is a well-documented phenomenon, citing [1-3] immediately when the topic is introduced (lines 051 - 053 of the revised manuscript).
> - **Proposition 1**: We have reframed this proposition to clarify that it summarizes established empirical findings from [1-3], rather than presenting a new theoretical result (lines 138 - 146 of the revised manuscript).
> - **Clarification of Contribution**: We have sharpened the text to define our specific contribution: a mathematically grounded generalization of Integrated Gradients (CDIG) that solves this specific, known limitation through invertible transforms (lines 053 - 056, 057 - 062 and 118 - 123 of the  revised manuscript.)
> We believe these changes resolve the ambiguity that led to the reviewer’s concern and make the provenance of the motivating observation explicit.
>
> **Section 3.2 and overlap with prior synthetic examples**
>
> While Section 3.2 shares conceptual similarities with the synthetic tasks in [2] and [3], it serves a distinct and necessary theoretical function in our paper. Unlike prior work, we use this specific setup to derive a closed-form analytical link between our Frequency-Domain IG and the frequency response of convolutional filters (Eq. 7). This derivation connects the "black box" explanation to known signal processing theory, providing a theoretical sanity check that was previously absent. We have revised the text to acknowledge the similarity to [2, 3] while highlighting this distinct theoretical purpose.
>
> **Limitations of IG and of our method**
>
> Our goal in this work is intentionally narrow: we only address one specific limitation of IG in time-series settings, namely the domain-mismatch / latent-feature failure mode [1-3]. All other limitations of IG (e.g. dependence on the baseline, possible issues due to gradient saturation and noise, requirement of differentiability, and computational cost) propagate unchanged to our cross-domain variant.
>
> In our original submission we discuss these aspects, specifically in Remarks 1, 2 and 3 of Section 4. In the revised manuscript we make this relationship more explicit by:
> 1. Stating clearly that our contribution targets only the domain-mismatch limitation (Appendix L).
> 2. Adding a short summary in Appendix L highlighting how IG limitations transfer into our method and how available solutions can be used in Cross-domain IG.
>
> **Beyond frequency: other domains**
>
> Our theoretical construction is defined for an arbitrary invertible, differentiable transform, not only for the Fourier transform. The frequency-domain example in Section 4.2 is chosen because convolutional filters admit a natural and familiar frequency-response interpretation, which makes it convenient to analytically demonstrate the match between CDIG and the model behaviour. In the experiments we already instantiate Cross-domain IG in three distinct transform domains: Fourier, ICA, and STL decomposition, to illustrate how our method can be applied with different transforms. In the revision we have also added the Complex Cepstrum domain, further illustrating the range of our model’s applications.

---

> > ### Author Response · Authors · 2025-11-23
> >
> > **Domain knowledge requirement**
> >
> > Our design choice is deliberate: Cross-domain IG is intended as a tool for domain experts who already reason in terms of frequency bands, independent components, trend/seasonality, etc. Our goal is to allow experts to inspect attributions directly in the domains they routinely use to analyse signals. We see the requirement of specifying an explanation domain as a feature that enables the incorporation of prior knowledge, not as an attempt to fully automate the discovery of all possible latent representations.
> >
> > At the same time, we agree that if the chosen transform is inappropriate or poorly understood, explanations can be misleading. These risks are already discussed in the original submission in Appendix K and in the ethical considerations.
> >
> > Automatic discovery of suitable explanation domains is, in our view, an important but orthogonal research direction. Our contribution in this paper is to provide a principled, IG-based framework for generating saliency maps once a transform domain has been specified. Methods for automatically proposing or learning such domains can be built on top of this framework. In Appendix K of the original manuscript we mention automatic transform discovery as a limitation of the current CDIG and future work.
> >
> > **On the discussion of results**
> >
> > We have added an additional Discussion section, where we discuss our results. We also have a brief summary of our limitations. An extended version of the limitations can be found in Appendix L.
> >
> > **On the use of general ML models**
> >
> > Regarding the model class, the Cross-domain IG definition only requires that the model be differentiable with respect to its input. Under this condition, the axiomatic guarantees we prove hold for any model, not just CNNs. The closed-form derivation in Section 4.2 is specific to a simple CNN and serves to make connection between CDIG and filter frequency responses fully explicit, but the algorithm itself can be applied unchanged to other differentiable architectures such as RNNs or transformers for time series. In our Applications section specifically we use a Convolutional Neural Network with attention mechanism, a transformer and a foundation model.
> >
> > **Quantitative Evaluation**: We have expanded the experimental section to include a model-agnostic evaluation on the AudioMNIST dataset. We now provide quantitative comparisons (insertion/deletion scores) against FreqRISE [4], and Virtual Inspection Layer on top of Time-LRP, and Time-IG [5]. Our method achieves performance comparable to or exceeding these baselines while offering the unique flexibility of cross-domain analysis.
> >
> > **Answers to questions**
> >
> > **Q1.** The failure mode of "feature mismatch" (where importance lies in a latent feature, not a time point) is addressed by transforming the integration path into the relevant domain (e.g., ICA or STL). However, standard IG limitations (like the baseline problem) remain. We have clarified this distinction in the revised Limitations section.
> >
> > **Q2.** Our method applies to any differentiable model (RNNs, Transformers, CNNs). The axiomatic guarantees of IG (Completeness, Sensitivity) are preserved by our derivation regardless of the architecture, provided the model is differentiable. We demonstrate this on Transformers (TimesFM) in Section 5.3.
> >
> > **Q3.** In our framework, the transform T is not restricted to a single transform, e.g. only Fourier. In particular, T can be defined as a composition of maps: $T = T1 \circ T2 \circ T3 \cdots$
> >
> > As long as this composite map is invertible and differentiable, the cross-domain IG definition applies without modification and yields attributions in this combined transform space.
> >
> > CDIG can partly address interactions via composite transforms: by applying a second transform within the components of a first one, we can hierarchically see which sub-features (e.g., frequencies within a seasonal component) are most important. Explicit interaction saliency between independent transforms, however, would require higher-order extensions and is left for future work.
> >
> > [1] Explainable AI for time series classification: a review, taxonomy and research directions
> >
> > [2] Post-hoc saliency methods fail to capture latent feature importance in time series data
> >
> > [3] What about the latent space? The need for latent feature saliency detection in deep time series classification
> >
> > [4] FreqRISE: Explaining time series using frequency masking
> >
> > [5] Explainable AI for time series via virtual inspection layers
> >
> > [6] FLEXtime: Filterbank Learning to Explain Time Series
> >
> > [7] Audiomnist: Exploring explainable artificial intelligence for audio analysis on a simple benchmark
> >
> > [8] Time is not enough: Time-frequency based explanation for time-series black-box models

---

> > > ### Comment · Reviewer_YxUK · 2025-11-27
> > > **Answer to rebuttal**
> > >
> > > I thank the authors for their rebuttal, which addressed most of my concerns. However, I still believe the work's contribution is limited in novelty. Therefore, I will raise my score to 4, but not higher.

---

### Official Review · Reviewer_8oeg · 2025-11-03

**Soundness:** 2
**Presentation:** 3
**Contribution:** 3
**Rating:** 4
**Confidence:** 3

**Summary:**

The authors advance explainability for time series methods by innovating on the integrated gradient method from 2017. Their proposal is called "Cross-domain Integrated Gradients", based on the fact that their method works on any invertible transformation. The user of the XAI method can thereby choose a transformation of their choice based on which domain is most suitable for explanations. The authors demonstrate qualitative feasibility using the Fourier Transform, Independent Component Analysis, and Seasonal-Trend decomposition.

**Strengths:**

1. The paper tackles an important and timely problem of developing XAI methods for time series. This field is in its infancy and underdeveloped, with the majority of XAI methods being developed on image data.

1. The authors provide both a TensorFlow and a PyTorch open-source library for their method.

1. Strong mathematical foundation for their method.

1. The method works, based on Figure 2-4, and is tested using three different transformations, in three different data domains.

**Weaknesses:**

As far as I can see, the paper has only one major weakness, which is its positioning in the existing state of the art. If the authors can address this, I will definitely consider changing my recommendation.

References to relevant prior literature on time series explainability are lacking in section 2, e.g, there are only two papers from 2024 in related work.

The experimental section is quite limited and is primarily qualitative. The only quantitative experiments I could find are in Appendix F, where the authors present insertion/deletion and compare that to time IG. There should be comparisons included with existing time-frequency analysis XAI methods. Comparisons should be made e.g to https://proceedings.mlr.press/v265/brusch25a.html, which also only assumes invertible transformations and therefore seems relevant.

**Questions:**

1. Equation 1:  $\xi$ can easily become negative; is that an issue?

1. Figure 1: Why is the peak of the orange distribution not located at 4 Hz, as specified by Eq. 1?

1. Equation 2: I'm a bit unsure about the notation in the integral with $x
+t(x-\hat{x})$, does this mean the partial derivative of $f$ is evaluated at that point, and then you integrate over $t$ ?

1. Line 352: $x(t) = a_1 \cos(2\pi · \xi_{hr}\cdot  t + \phi) + a2 \cos(2\cdot \pi(2\xi_{hr})\cdot  t + \phi))$. Small esthetic typo: should it be $2\pi$ without the cdot in the second cosine?

1. Figure 2: The blue dashed (as opposed to solid) lines make this figure more difficult to read, but I may be wrong.

1. Line 389: What does the $\rightarrow$ mean in this context?

---

> ### Author Response · Authors · 2025-11-23
>
> We thank the reviewer for highlighting the need to better position our work within the existing literature and for the constructive suggestion to include quantitative comparisons. We have expanded Section 2 - related works to include recent and relevant studies on time series explainability, specifically [1-6]. These additions provide a more comprehensive view of the current landscape.
>
> In our original submission, we provided a theoretical proof that Cross-domain IG is equivalent to Virtual Inspection Layers (VIL) [1] when applied to the frequency domain (DFT) by comparing the saliency formula derivation of both methods. Based on the constructive suggestions from the reviewers to include quantitative empirical evidence, we have performed new simulations on the AudioMNIST dataset [7]. We compared Cross-domain IG against FreqRISE [2], LRP + VIL, IG + VIL generating frequency domain attributions. Empirically, our method yields identical results to IG + VIL in the frequency domain, validating our theoretical derivation. When compared to FreqRISE, our method achieves comparable performance. Crucially, while existing methods (like FreqRISE and VIL) are restricted to the frequency domain, our method is domain-agnostic. To demonstrate this unique advantage, we provide the first-ever saliency results in the Complex Cepstrum domain within the same experimental table. This establishes a new benchmark for cross-domain saliency methods.
>
> **Answer to questions**
>
> **Q1.** No, this is not an issue. In the signal model $x(t) = \cos(2\pi\xi t + \phi)$, a negative frequency $\xi$ results in a phase shift. Since the phase $\phi$ is a free parameter in our formulation, negative values of $\xi$ do not invalidate the model.
>
> **Q2.** We confirm that the peak of the orange distribution in Figure 1 is indeed centered at exactly 4 Hz. The visual perception might be affected by the x-axis scaling or kernel density estimation smoothing, but the underlying data generation strictly follows Equation 1.
>
> **Q3.** Yes, your interpretation is correct. We evaluate the gradient at the interpolated point along the path. Formally, $\frac{\partial f}{\partial x_i}\bigg|_{\boldsymbol{x}^{\prime} + t \cdot(\boldsymbol{x} - \hat{\boldsymbol{x}})}$ calculates the partial derivative of the model f with respect to the input at the point $\boldsymbol{x}^{\prime} + t \cdot(\boldsymbol{x} - \hat{\boldsymbol{x}})$.
>
> **Q4.** We thank the reviewer for catching this; we have removed the erroneous dot in the second cosine term.
>
> **Q5.** We appreciate the feedback on legibility. The intent of the dashed vertical lines in Figure 1(a) is to indicate the specific samples drawn from the two distributions. These specific samples are then visualized in the Time Domain in Figure 1(b) and the Frequency Domain in Figure 1(c) to illustrate the transformation. We have revised the caption to clarify this mapping and adjust the line styles to improve readability.
>
> **Q6.** The arrow symbol $(\rightarrow)$ in parentheses is a legend reference indicating that "external interferences" are marked by black arrows in the corresponding figure. We have rephrased this sentence in the revision to make the reference explicit and clear.
>
> [1] Explainable AI for time series via virtual inspection layers
>
> [2] FreqRISE: Explaining time series using frequency masking
>
> [3] Post-hoc saliency methods fail to capture latent feature importance in time series data
>
> [4] What about the latent space? The need for latent feature saliency detection in deep time series classification
>
> [5] Explainable AI for time series classification: a review, taxonomy and research directions
>
> [6] FLEXtime: Filterbank Learning to Explain Time Series
>
> [7] Audiomnist: Exploring explainable artificial intelligence for audio analysis on a simple benchmark

---

### Meta-Review · Area_Chair_BYLJ · 2026-01-02

**Summary:**

The reviewers' concerns center on a few key points. (1) Novelty: Multiple reviewers note opportunities to clarify how this proposed work fits into the literature, along with a lack of sufficient citations. The authors largely resolve this issue, though the paper will continue to improve with a substantial revision of the experiments that incorporate direct comparisons where possible. Reviewers also struggled to understand when/why the new method is expected to outperform others. While the authors' response to R4 outlines some key differences, these should be clearly emphasized and experimented with in order to make these claims (these differences don't seem to hinder direct comparison to prior works). (2) Experimental Sufficiency: Multiple reviewers expressed concerns that the experiments over-emphasize case studies/examples, and omit quantitative measures and direct comparisons with prior works. While some of this information is in the Appendix, the experiments section will need substantial rewriting.

**Reviewer Concerns:**

**Addressed Concerns**:
* Attributions to prior works
* Describing the domain knowledge requirements in the limitations will help
* Computing insertion/deletion scores (partially) --- while these exist, they should be expanded where possible and moved into the main paper to inform the main conclusions

**Outstanding Concerns**:
* Challenges to overcome could be more-clearly stated, possibly in the related works or preliminaries
* Quantitative metrics should appear in the main paper and drive the main conclusions

**Reviewer Scores:**

R1: Slight increase

R2: Slight increase

R3: Unchanged

R4: Unchanged

---

### Decision · Program_Chairs · 2026-01-26

Reject